# The impact of local genomic properties on the evolutionary fate of genes

Yuichiro Hara[1]*, Shigehiro Kuraku[2,3,4]

[1]Research Center for Genome & Medical Sciences, Tokyo Metropolitan Institute of Medical Science, Tokyo, Japan; [2]Molecular Life History Laboratory, Department of Genomics and Evolutionary Biology, National Institute of Genetics, Mishima, Japan; [3]Department of Genetics, Sokendai (Graduate University for Advanced Studies), Mishima, Japan; [4]RIKEN Center for Biosystems Dynamics Research, Kobe, Japan

**Abstract** Functionally indispensable genes are likely to be retained and otherwise to be lost during evolution. This evolutionary fate of a gene can also be affected by factors independent of gene dispensability, including the mutability of genomic positions, but such features have not been examined well. To uncover the genomic features associated with gene loss, we investigated the characteristics of genomic regions where genes have been independently lost in multiple lineages. With a comprehensive scan of gene phylogenies of vertebrates with a careful inspection of evolutionary gene losses, we identified 813 human genes whose orthologs were lost in multiple mammalian lineages: designated 'elusive genes.' These elusive genes were located in genomic regions with rapid nucleotide substitution, high GC content, and high gene density. A comparison of the orthologous regions of such elusive genes across vertebrates revealed that these features had been established before the radiation of the extant vertebrates approximately 500 million years ago. The association of human elusive genes with transcriptomic and epigenomic characteristics illuminated that the genomic regions containing such genes were subject to repressive transcriptional regulation. Thus, the heterogeneous genomic features driving gene fates toward loss have been in place and may sometimes have relaxed the functional indispensability of such genes. This study sheds light on the complex interplay between gene function and local genomic properties in shaping gene evolution that has persisted since the vertebrate ancestor.

*For correspondence:
hara-yi@igakuken.or.jp

## Editor's evaluation

The study provides a fundamental understanding of the driving forces behind gene losses in genome evolution and connects the propensity for gene losses to local genomic features like mutation rate and expression pattern. The methodology is compelling, as it identifies "elusive human genes" through independent gene losses in at least two mammalian lineages. The comparative genomics and statistical analyses are thorough and rigorous, making this study appealing to readers interested in exploring the global patterns and underlying mechanisms of gene fate evolution across the phylogenetic tree.

## Introduction

In the course of evolution, genomes continue to retain most genes with occasional duplications, while losing some genes (**Blomme et al., 2006**; **Fernández and Gabaldón, 2020**; **Shen et al., 2018**). This retention and loss can be interpreted as gene fate; genes are stably retained in the genome, but some factors may cause them to transition to a state where deletion occurs. Accordingly, identification of the factors allowing gene loss may facilitate our understanding of gene fate. Gene retention or loss

has generally been considered to depend largely on the functional importance of the particular gene from the perspective of molecular evolutionary biology (*Albalat and Cañestro, 2016*; *Bartha et al., 2018*; *Blanc et al., 2012*; *Liu et al., 2015*; *Olson, 1999*; *Sharma et al., 2018*; *Shen et al., 2018*). Genes with indispensable functions have usually been retained with highly conserved sequences in genomes through rapid elimination of alleles that impair gene functions (*Hirsh and Fraser, 2001*; *Krylov et al., 2003*; *Miyata et al., 1980*; *Pál et al., 2006*). On the contrary, genes with less important functions are likely to accept more mutations and structural variations, which can degrade the original functions, leading to gene loss through pseudogenization or genomic deletion (*Jordan et al., 2002*; *Yang et al., 2003*). To date, gene loss has been imputed to the relaxation of functional constraints of individual genes. Gene loss has further been revealed to drive phenotypic adaptation in various organisms (*Albalat and Cañestro, 2016*; *Olson, 1999*), as well as in a gene knockout collection of yeasts in culture (*Giaever and Nislow, 2014*; *Maclean et al., 2017*).

To uncover the association between fates and functional importance of the genes, molecular evolutionary analyses have been conducted at various scales, from gene-by-gene to genome-wide. A number of studies have revealed that the genes with reduced non-synonymous substitution rates (or $K_A$ values) and ratios of non-synonymous to synonymous substitution rates ($K_A/K_S$ ratios) are less likely to be lost (*Jordan et al., 2002*; *Yang et al., 2003*). A genome-wide comparison of duplicated genes in yeast revealed larger $K_A$ values for those lost in multiple lineages than those retained by all the species investigated (*Byrne and Wolfe, 2007*). Other comprehensive studies of gene loss across metazoans and teleosts revealed that the genes expressed in the central nervous system are less prone to loss (*Fernández and Gabaldón, 2020*; *Roux et al., 2017*). These observations again suggest that gene fate depends on the functional constraints of a particular gene.

Besides functional constraints, several studies have identified the genes lost independently in multiple lineages, revealing that the genomic regions containing these genes 'prefer' particular characteristics associated with structural instability (*Cortez et al., 2014*; *Hughes et al., 2012*; *Lewin et al., 2021*; *Maeso et al., 2016*). In mammals, tandemly arrayed homeobox genes derived from the Crx gene family were lost in multiple species (*Lewin et al., 2021*; *Maeso et al., 2016*). The findings suggest that genomic features containing tandem duplications facilitate unequal crossing over, leading to frequent gene loss. Mammalian chromosome Y, which contains abundant repetitive elements and continues to reduce in size, has lost a considerable number of genes (*Cortez et al., 2014*; *Hughes et al., 2012*). In the stickleback genome, a *Pitx1* enhancer was independently lost in multiple lineages inhabiting freshwater due to its genomic location in a structurally fragile site, leading to recurrent loss of pelvic fins (*Xie et al., 2019*). Genes and genomic elements in such particular regions may be prone to loss in a more neutral manner than the relaxation of functional importance or via functional adaptations. Accordingly, these studies focusing on the particular genomic regions led us to search for the common features in genomes that potentially facilitate gene loss. Genome-wide scans have revealed heterogeneous distributions of a variety of sequence and structural features so far, for example, base composition (*Bernardi and Bernardi, 1986*; *Cohen et al., 2005*; *Katzman et al., 2011*), the frequency of repetitive elements (*Korenberg and Rykowski, 1988*; *Medstrand et al., 2002*), and DNA-damage sensitivity induced by replication inhibitors (*Debatisse et al., 2012*; *Helmrich et al., 2006*). However, the extent to which these characteristics are associated with gene fates has not been understood well at a genome-wide level.

The accumulation of near-complete genome assemblies for various organisms facilitates comprehensive taxon-wide analysis of gene loss (*Fernández and Gabaldón, 2020*; *Guijarro-Clarke et al., 2020*; *Rice and McLysaght, 2017*). Along with this motivation, we recently performed a comprehensive analysis on the fate of paralogs generated via the two-round whole-genome duplications in early vertebrates (*Hara et al., 2018a*). The results revealed that the genes retained by reptiles but lost in mammals and Aves rapidly accumulated not only non-synonymous but also synonymous substitutions in comparison with the counterparts retained by almost all the vertebrates examined, indicating that those genes prone to loss show increasing mutation rates. Furthermore, these loss-prone genes were located in genomic regions with high GC contents, high gene densities, and high repetitive element frequencies. These findings suggest that the fates of those genes are influenced not only by functional constraints but also by intrinsic genomic characteristics. Because the findings were restricted to a set of particular genes, they prompted us to examine whether this trend is associated with gene fates on a genome-wide scale.

In this study, we inferred molecular phylogenies of vertebrate orthologs to systematically search for the genes harboring different fates in the human genome. We previously referred to the nature of genes prone to loss as 'elusive' (*Hara et al., 2018a*; *Hara et al., 2018b*). In this study, we define the elusive genes as those that are retained by modern humans but have been lost independently in multiple mammalian lineages. As a comparison of the elusive genes, we retrieved the genes that were retained by almost all of the mammalian species examined and defined them as 'non-elusive,' representing those persistent in the genomes. We conducted a careful search for gene loss to reduce the false discovery rate (FDR), which is usually caused by incomplete sequence information (*Botero-Castro et al., 2017*; *Deutekom et al., 2019*). By comparing the genomic regions containing these genes, we uncovered genomic characteristics relevant to gene loss. We associated the elusive genes with a variety of findings from deep sequencing analyses of the human genome, including transcriptomics, epigenomics, and genetic variations. These data assisted us to understand how intrinsic genomic features may affect gene fate, leading to gene loss by decreasing the expression level and eventually relaxing the functional importance of 'elusive' genes.

## Results

### Identification of human 'elusive' genes

We defined an 'elusive' gene as a human protein-coding gene that existed in the common mammalian ancestors but was lost independently in multiple mammalian lineages (*Figure 1*; see 'Materials and methods' for details). We searched for such genes by reconstructing phylogenetic trees of vertebrate orthologs and detecting gene loss events within the individual trees. To search for elusive genes, we paid close attention to distinguishing true evolutionary gene loss from falsely inferred gene loss caused by insufficient genome assembly, gene prediction, and orthologous clustering (*Botero-Castro et al., 2017*; *Deutekom et al., 2019*), as described below.

We first produced highly complete orthologous groups comprised of nearly complete gene sets. We merged multiple gene annotations of a single species followed by assessments of the completeness of the gene sets (*Figure 1a*). Using these gene sets, we then created two sets of ortholog groups with different methods and merged them into a single set (*Figure 1a*). In searching for gene loss events, we restricted our study to those that occurred in the common ancestors of particular taxonomic groups. This procedure relieved false identifications of gene loss in a species or an ancestor of a lower taxonomic hierarchy caused by incomplete genomic information (*Figure 1b*).

We integrated gene annotations from Ensembl, RefSeq, and the sequence repositories of individual genome sequencing projects to produce gene annotations for 114 mammalian and 132 non-mammalian vertebrates. From these, we selected the annotations of 101 and 90 species, respectively, that exhibited high completeness in the BUSCO assessment (*Simão et al., 2015*; Supplementary Table S1 in *Supplementary file 1a*). Using these gene sets, clustering of ortholog groups was conducted by OrthoFinder, and these groups were integrated into the ortholog groups provided by the Ensembl Gene Tree. This integration resulted in 50,768 vertebrate ortholog groups. Phylogenetic tree inference of the integrated ortholog groups and pruning of the individual trees based on gene duplications resulted in 17,495 mammalian ortholog groups that contained human genes. We classified the mammalian species into 15 taxonomic groups ranging from order to family (listed in Table S1; *Supplementary file 1a*). For the individual mammalian orthologs, we searched for the taxa in which the gene was absent in all the species examined (*Figure 1b*). We interpreted this gene absence as an evolutionary loss that occurred in the common ancestor of the taxon. Validating the gene loss through an ortholog search in genome assemblies and synteny-based ortholog annotations, we extracted the ortholog groups that were retained by humans but were lost independently in the common ancestors of at least two taxa (*Figure 1c*). Hereafter we call the human genes belonging to these ortholog groups 'elusive genes.' To compare these, we also selected the ortholog groups that contained all of the mammals examined including single-copy human genes. We called these 'non-elusive genes.' This comprehensive scan of gene phylogenies resulted in 813 elusive and 8050 non-elusive genes (Supplementary Table S2; *Supplementary file 2*).

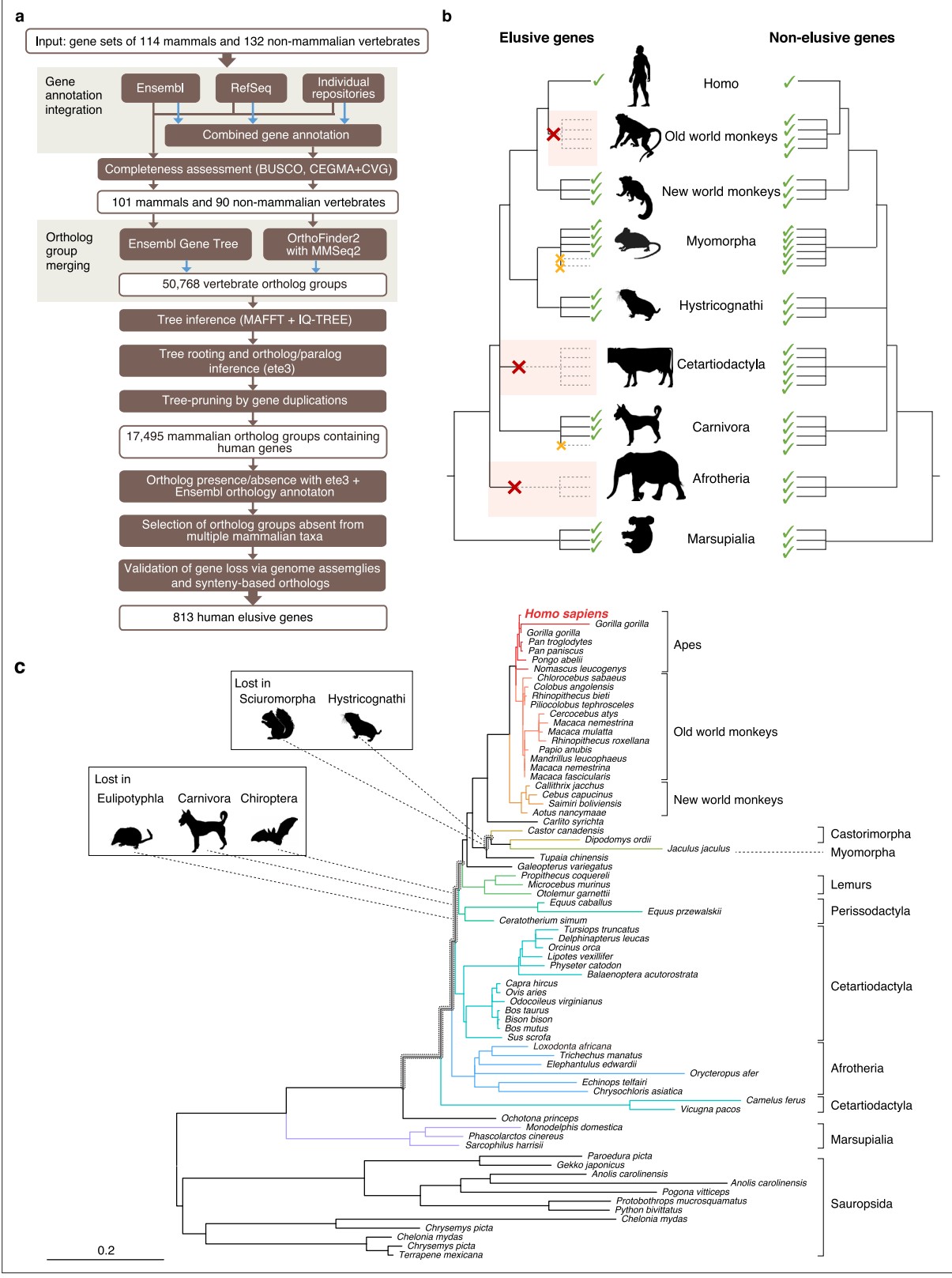

**Figure 1.** Detection of 'elusive' genes. (**a**) Pipeline of ortholog group clustering and gene loss detection. (**b**) Definition of an elusive gene schematized with ortholog presence/absence pattern referring to a taxonomic hierarchy. Red and orange crosses denote the gene loss in the common ancestor of a taxon and the loss specific to a single species, respectively. (**c**) A representative phylogeny of the elusive gene encoding Chitinase 3-like 2 (CHI3L2). Taxa shown in the tree were used to investigate the presence or absence of orthologs. The Sciuromorpha, Hystricognathi, Eulipotyphla, Carnivora,

*Figure 1 continued on next page*

and Chiroptera are absent from the tree, indicating that the CHI3L2 orthologs were lost somewhere along the branches framed in gray in the tree. In addition, the orthologs of many members of the Myomorpha were not found, suggesting that gene loss occurred in this lineage.

## Genomic signatures of the human elusive genes

The loss-prone nature of the elusive genes suggests a relaxation of their functional constraints. To uncover the molecular evolutionary characteristics associated with each elusive gene, we computed synonymous and non-synonymous substitution rates in coding regions, namely $K_S$ and $K_A$, respectively, between human and chimpanzee and mouse orthologs for the elusive and non-elusive genes. In addition, we computed nucleotide substitution rates for introns ($K_I$) between human and chimpanzee (*Pan troglodytes*) orthologs and compared them between the elusive and non-elusive genes. The results showed larger $K_A$ values in the ortholog pairs of the elusive genes than in those of the non-elusive genes (*Figure 2a*, *Figure 2—figure supplement 1*). This indicates a rapid accumulation of amino acid substitutions in the elusive genes, potentially accompanied by the relaxation of functional constraints. Our analysis further illuminated larger $K_S$ and $K_I$ values for the elusive genes than in the non-elusive

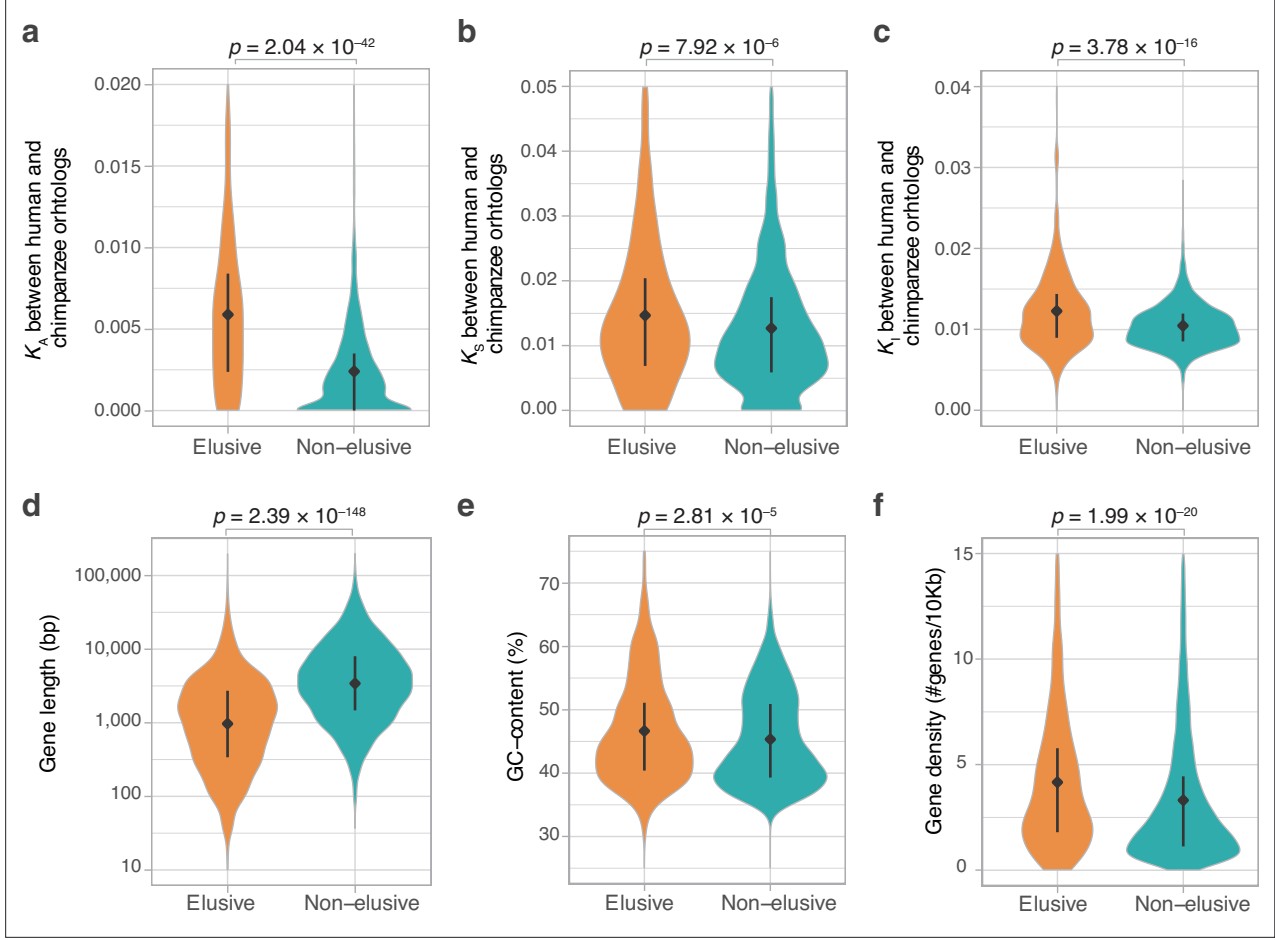

**Figure 2.** Genomic and evolutionary characteristics of elusive genes. Distributions of non-synonymous, synonymous, and intronic nucleotide substitution rates, namely $K_A$ (**a**), $K_S$ (**b**), and $K_I$ (**c**) values, respectively, between the human–chimpanzee orthologs of the elusive and non-elusive genes. Distribution of gene length (**d**) and GC content (**e**) of the human elusive and non-elusive genes. (**f**) Distribution of gene density in the genomic regions where the human elusive and non-elusive genes are located. The plots consist of 249 elusive and 5145 non-elusive genes that retained chimpanzee orthologs (**a, b**), 473 and 4626 of those which harbored introns aligned with the chimpanzee genome (**c**; see 'Materials and methods'), and all of the 813 elusive and 8050 non-elusive genes (**d–f**). Diamonds and bars within violin plots indicate the median and range from the 25th to 75th percentile, respectively.

The online version of this article includes the following figure supplement(s) for figure 2:

**Figure supplement 1.** Comparison of $K_A$ and $K_S$ values between orthologs of the elusive and non-elusive genes.

genes (*Figure 2b and c*, *Figure 2—figure supplement 1*). Importantly, the higher rate of synonymous and intronic nucleotide substitutions, which may not affect changes in amino acid residues, indicates that the elusive genes are also susceptible to genomic characteristics independent of selective constraints on gene functions.

To further scrutinize the characteristics reflecting the genomic environment rather than gene function, we analyzed genomic characteristics that may distinguish the elusive from non-elusive genes. A comparison between these two categories revealed shorter gene-body lengths and higher GC contents of elusive rather than non-elusive genes (*Figure 2d and e*). Furthermore, a scan of intragenomic gene distribution revealed that the elusive genes were located in the genomic regions with high gene density compared with the non-elusive genes (*Figure 2f*). Our findings indicate that such elusive genes have distinct characteristics in the human genome. These genomic characteristics, as well as high nucleotide substitution rates, were consistent with the findings in our genome analyses using the amniote and elasmobranch genomes (*Hara et al., 2018a*; *Hara et al., 2018b*).

## Tracing elusiveness back along the vertebrate evolutionary tree

The origins of the human elusive genes can be traced back along the evolutionary tree, at least to the mammalian common ancestor. To investigate possible antiquities of the genomic properties associated with elusive genes, we investigated their orthologs in non-mammalian vertebrates by scrutinizing the ortholog groups used for elusive gene identification. We found that 152 out of 813 elusive genes originated in mammalian lineages, and this proportion was larger than those of the elusive genes (65 out of 8050, p=2.50 × 10^{-110}), indicating that the elusive genes are more abundant in recently born genes than non-elusive genes. We then selected 517 elusive and 7900 non-elusive genes that originated in the common ancestors of jawed vertebrates or earlier. These

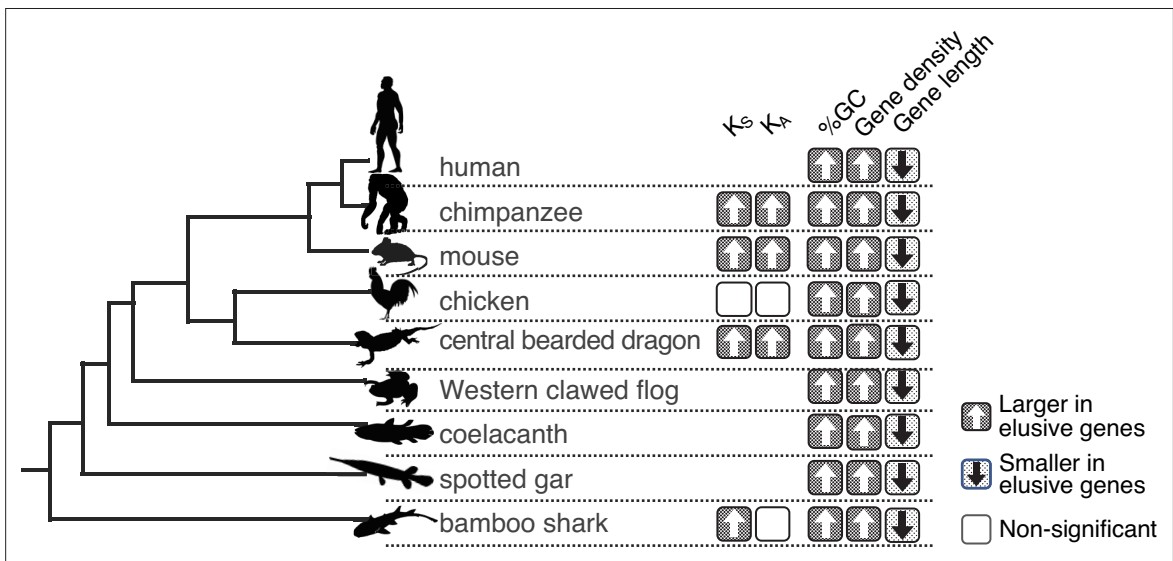

**Figure 3.** Long-standing characteristics of elusive genes. Retention of the genomic and evolutionary characteristics of the human elusive genes across vertebrates. The individual round squares with arrows indicate significant increases or decreases of the distribution of particular characteristics in the orthologs of the human elusive genes and their flanking regions compared with those of the non-elusive genes in these selected vertebrate genomes. For the chimpanzee and mouse genomes, $K_A$ and $K_S$ values were computed between the human elusive genes and the orthologs of these mammals. For non-mammalian species, these values were computed with ortholog pairs for the elusive/non-elusive genes between the corresponding species and their closely related species: turkey for chicken, green anole for central bearded dragon, and whale shark for bamboo shark. Distributions of these metrics for non-human species are shown in *Figure 2—figure supplement 1* and *Figure 3—figure supplement 2*. Species name: mouse, *Mus musculus*; chicken, *Gallus gallus*; central bearded dragon, *Pogona vitticeps*; Western clawed frog, *Xenopus tropicalis*; coelacanth, *Latimeria chalumnae*; spotted gar, *Lepisosteus oculatus*; bamboo shark, *Chiloscyllium plagiosum*.

The online version of this article includes the following source data and figure supplement(s) for figure 3:

**Figure supplement 1.** Asymmetric ortholog retention across the vertebrates.

**Figure supplement 1—source data 1.** A 2×2 contingency table in Figure 3—figure supplement 1.

**Figure supplement 2.** Genomic characteristics of the orthologs of elusive and non-elusive genes.

subsets allowed us to examine the degree of retention of non-mammalian vertebrate orthologs in the elusive and non-elusive genes. On average, approximately 40% of these elusive genes were found to be retained by non-mammalian vertebrates, while this proportion increased up to 90% for the non-elusive genes. (*Figure 3—figure supplement 1a*). In the coelacanth, gar, and shark, the orthologs of the elusive genes were less frequently retained by all the species than those of the non-elusive ones (*Figure 3—figure supplement 1b*). The results suggest that the origins of the loss-prone propensity of the elusive genes potentially date back to the period long before the emergence of the Mammalia.

We further examined the genomic characteristics associated with the human elusive genes in the vertebrate orthologs. In all the species examined, orthologs of the elusive genes exhibited high GC content and compact gene bodies. Additionally, in most of these species, the orthologs of elusive genes were located in genomic regions with high gene density compared with orthologs of the non-elusive genes (*Figure 3*, *Figure 3—figure supplement 2*). In addition, we computed $K_S$ and $K_A$ values between the orthologs of the vertebrate species and their close relatives for elusive and non-elusive genes. In any of the species pairs except for avians, the orthologs of the elusive genes were found to harbor higher $K_A$ and $K_S$ values than those of the non-elusive gene orthologs (*Figure 3*, *Figure 2—figure supplement 1*). These observations indicate that these genomic characteristics probably originated before the emergence of gnathostomes, a monophyletic group of chondrichthyan and bony vertebrates, and have been retained for approximately 500 million years.

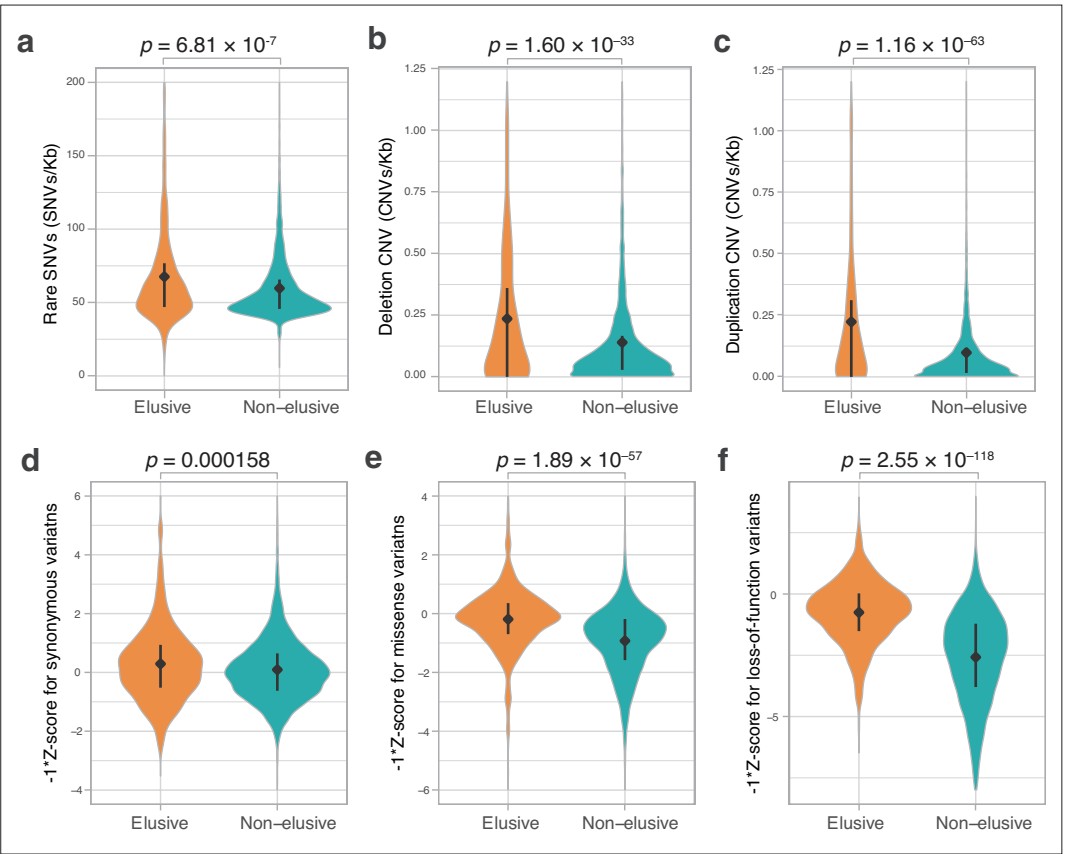

**Figure 4.** Genetic variations of the elusive and non-elusive genes within human populations. Comparison of the density of rare single-nucleotide variants (SNVs) (**a**), deletion copy number variants (CNVs) (**b**), duplication CNVs (**c**), and Z-scores of synonymous (**d**), missense (**e**), and loss-of-function variants (**f**). We used opposite numbers of the Z-scores in **d–f** so that the elusive genes have higher values than non-elusive genes as in *Figure 2a, b, c, e, f* and *Figure 3a–c*. (**a–c**) 813 elusive genes and 8050 non-elusive genes were used. (**d–f**) 544 elusive genes and 7303 non-elusive genes for which genetic variants were available in GnomAD were used. Diamonds and bars within violin plots indicate the median and range from 25th to 75th percentile, respectively.

## Abundant polymorphism in elusive genes

The observation of large $K_S$ and $K_A$ values in the elusive genes prompted us to examine the extent to which these genes have accommodated genetic variations in modern humans. Large-scale human genome resequencing projects have identified a huge number of genetic variations, from rare to common, and from single-nucleotide variants (SNVs) to chromosome-scale structural variants, facilitating tackling this issue. We retrieved copy number variants (CNVs) and rare SNVs in the human genome from the Database of Genomic Variants, release 2016-08-31 (*MacDonald et al., 2014*) and dbSNP release 147 (*Sherry et al., 2001*), respectively, and computed their densities in the individual genic regions. We found that the genic regions of the human elusive genes contained abundant rare SNVs, as well as deletion and duplication CNVs, compared with those of the non-elusive genes (*Figure 4a–c*). This result suggests that genomic regions containing the elusive genes are not only prone to loss but also to duplication.

To evaluate the functional consequences of abundant genetic variants in the elusive genes, we investigated genetic variations stored in the gnomAD v. 2.1 database, a repository containing >120,000 exome and >15,000 whole-genome sequences of human individuals (*Karczewski et al., 2021*). This database classifies SNVs in coding regions into three categories—synonymous, missense, and loss-of-function—and the loss-of-function category contains nonsense mutations, frameshift mutations, and mutations in splicing junctions. The gnomAD site computes a $Z$-score, an index representing the abundance of SNVs for individual genes; positive and negative values denote fewer or more mutations in a coding region than expected, respectively (*Figure 4d–f*). Accordingly, the $Z$-score for nonsense mutations and loss-of-function mutations of the individual genes indicates the degree of natural selection: larger values demonstrate genes subjected to purifying selection, while smaller ones suggest functional relaxation. We found lower $Z$-scores of missense and loss-of-function mutations (higher opposite numbers of $Z$-scores in *Figure 4e and f*) in the human elusive genes than in the non-elusive genes, suggesting that the elusive genes are more functionally dispensable and potentially tolerable to harmful mutations. Additionally, opposite numbers of $Z$-scores of synonymous mutations of the human elusive genes were higher than those of the non-elusive genes (*Figure 4d*). This confirms the high mutability of genomic regions containing elusive genes, as observed in the $K_S$ values.

## Transcriptomic natures of elusive genes

To further investigate how the human elusive genes have decreased functional essentiality, we examined their expression profiles. For this purpose, we compared gene expression profiles of the 54 adult tissues from the GTEx database v. 8 (*The GTEx Consortium et al., 2020*) between the elusive and non-elusive genes. For individual genes, we computed the maximum transcription per million (TPM) values among these tissues as the expression quantity level. For expression diversities, we employed Shannon's diversity index $H'$, which is often utilized as an index of species diversity in the ecological literature, based on the proportion of TPM values across the 54 tissues.

As shown in the density scatter plots of the individual genes displaying these two indicators in *Figure 5*, most of the non-elusive genes possessed large maximum TPM and $H'$ values. Thus, most non-elusive genes are ubiquitously expressed at certain levels. By contrast, the density plot of the elusive genes displayed an additional high-density spot with small TPM and $H'$ values, indicating that the genes in this spot were not expressed, at least in adult tissues. The plot also showed another broad dense area of small $H'$ values, which contained the genes expressed in a single or a few tissues. A similar analysis was performed with the fetal single cell RNA-seq data (*Cao et al., 2020*), revealing that the averaged expression profiles of the elusive and non-elusive genes for the 172 cell types were concordant with those of the adult tissues (*Figure 5*). Our findings demonstrate that some elusive genes harbor low-level and spatially restricted expression profiles, that is, less pleiotropic states, which are rarely observed in the non-elusive genes.

## Epigenetic nature of elusive genes

Our finding of the low-level and spatially restricted expression patterns of elusive genes prompted us to explore epigenetic properties involved in this transcriptional regulation. Therefore, we retrieved epigenetic data on a variety of human cell lines from a few regulatory genome databases including ENCODE, a repository that stores the comprehensive annotations of functional elements in the human

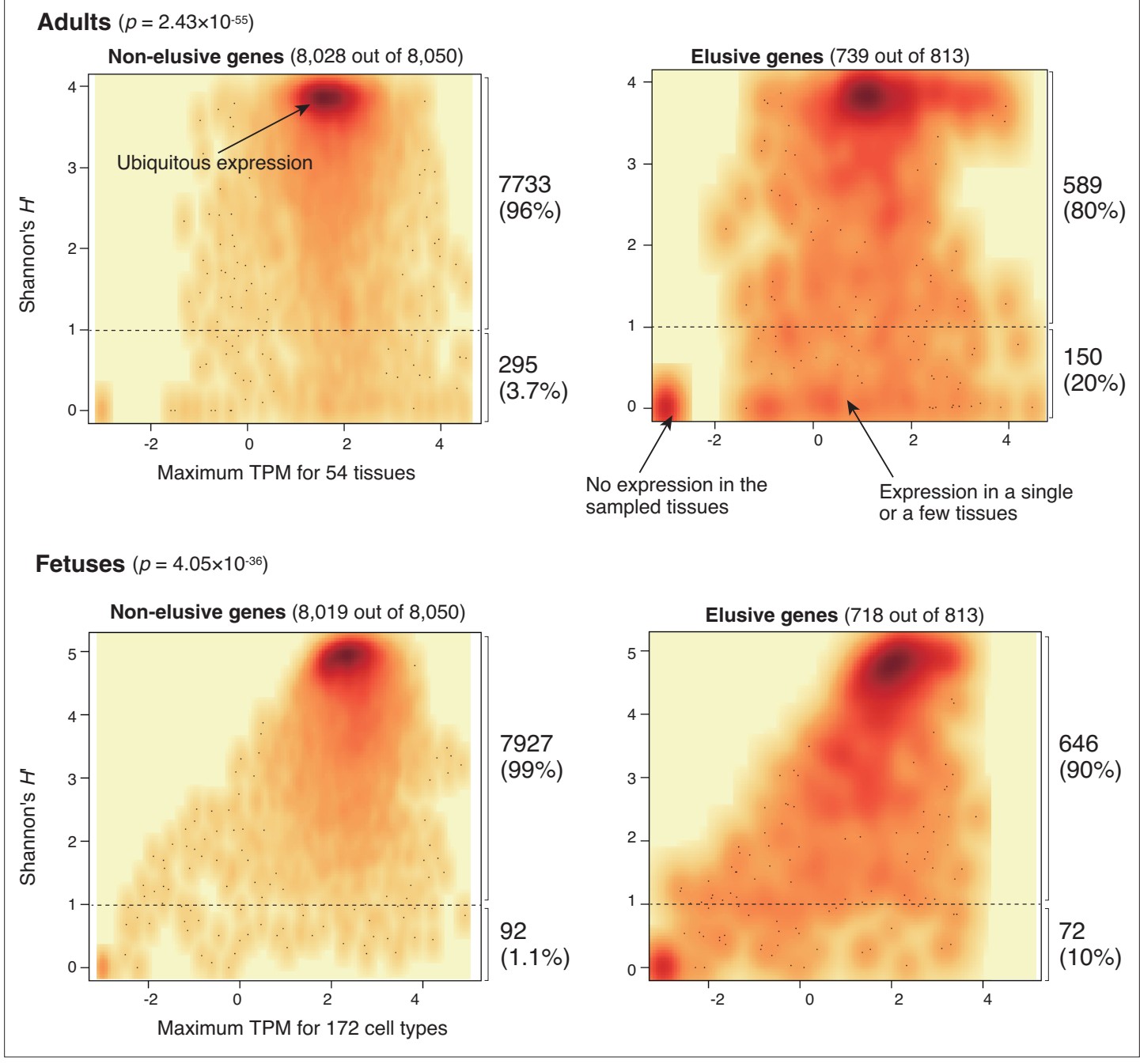

**Figure 5.** Expression profiles of elusive and non-elusive genes. The figure shows density scatter plots of the expression quantity and divergence of elusive and non-elusive genes. The numbers of the elusive/non-elusive genes and those for which the expression quantities were available are indicated in each panel. p-values were computed via 2 × 2 contingency tables presenting numbers of elusive and non-elusive genes with $H' < 1$ and $H' \geq 1$. The median transcription per million (TPM) value of each of the adult tissue across individuals was retrieved from the GTEx database (**The GTEx Consortium et al., 2020**), and normalized TPM values of the fetal cell types were retrieved from the Descartes database (**Cao et al., 2020**). For the individual genes, maximum TPM and Shannon's $H'$ values were computed using these processed TPM values.

The online version of this article includes the following figure supplement(s) for figure 5:

**Figure supplement 1.** Expression profiles of the orthologs of the elusive and non-elusive genes for non-mammalian vertebrates.

genome (**The ENCODE Project Consortium, 2012**). Using this information, we characterized the epigenetic features of the genomic regions containing elusive genes (**Figure 6**).

We compared peak densities based on the Assay for Transposase-Accessible Chromatin using sequencing (ATAC-seq), an indicator of accessible chromatin regions in the genome, in gene bodies

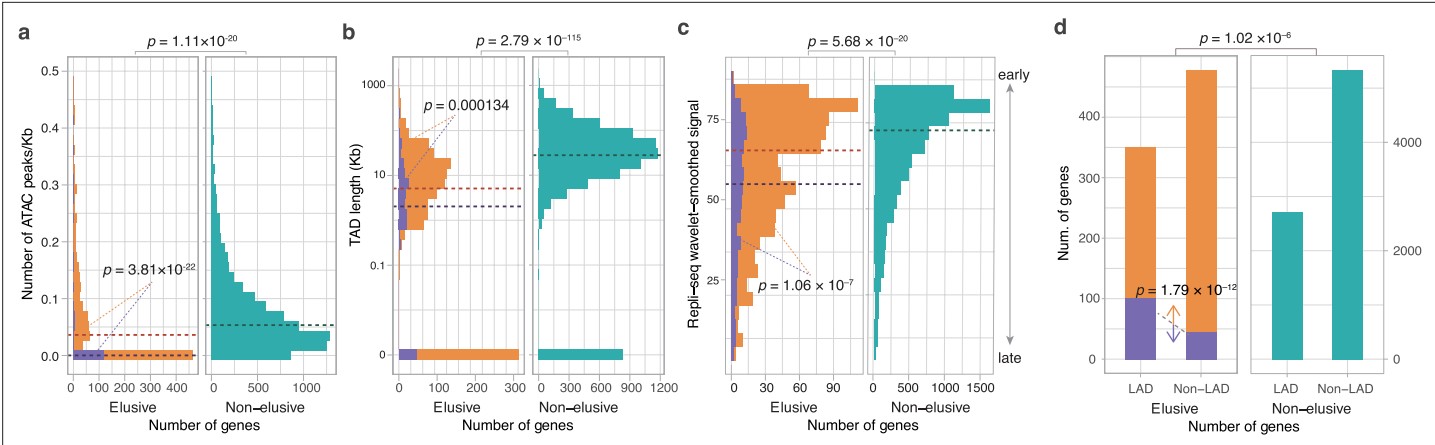

**Figure 6.** Epigenetic features of the elusive genes. Comparison of the distribution of ATAC-seq peak density (**a**), length of the topologically associating domains (TADs) including the elusive or non-elusive genes (**b**), the replication timing indicator based on Repli-seq (**c**), and overlap with the lamina-associated domains (LADs) computed from Lamin B1 ChIP-seq data. All of the analyses were performed by using the processed sequencing data publicly available (Table S3; ***Supplementary file 1b***). (**d**) ATAC-seq and Hi-C were performed with A549 cells, Repli-seq was performed with HepG2 cells, and Lamin B1 ChIP-seq was performed with HAP-1 cells. In the elusive gene panels (orange bar), purple bar indicates the elusive genes with restricted expressions ($H' < 1$; ***Figure 5***). p-values for individual panels indicate the comparison between the elusive (813) and non-elusive (8050) genes and the one between the elusive genes with $H' < 1$ (150) and those with $H' \geq 1$ (589). The results for other cells are shown in ***Figure 6—figure supplements 1–4*** For the individual epigenetic characteristics, correction for multiple testing was performed for comparison in each cell cultures.

The online version of this article includes the following figure supplement(s) for figure 6:

**Figure supplement 1.** ATAC-seq peak density of the elusive and non-elusive gene regions.

**Figure supplement 2.** Sequence lengths of the topologically associating domains (TADs) containing elusive or non-elusive genes.

**Figure supplement 3.** Comparison of the replication timing indicator based on Repli-seq between the elusive and non-elusive genes.

**Figure supplement 4.** The fraction of elusive and non-elusive genes that overlap with lamina-associated domains (LADs).

**Figure supplement 5.** ATAC-seq peak density of the chicken orthologs of the elusive and non-elusive gene regions.

and flanking regions between the elusive and non-elusive genes. In all of the eight cell lines examined (11 samples in total), the results showed fewer ATAC-seq peaks in the genomic regions including the elusive genes than in those including non-elusive genes, indicating that the elusive genes are likely to reside in inaccessible genomic regions (***Figure 6a***, ***Figure 6—figure supplement 1***). We also searched for topologically associating domains (TADs), genomic elements with frequent physical self-interaction potentially acting as promoter-enhancer contacts (***Rao et al., 2014***) that included either the elusive or non-elusive genes. The result showed that a higher fraction of the elusive genes resided outside of the TADs than the non-elusive genes for all the eleven cell lines investigated (***Figure 6b***, ***Figure 6—figure supplement 2***). Furthermore, the elusive genes were located in shorter TADs. These observations suggest that the elusive genes are unlikely to be regulated by distant regulatory elements compared with the non-elusive genes (***Figure 6b***, ***Figure 6—figure supplement 2***).

Our investigations extended to the association of the elusive genes with further global regulation of genomic structures. We compared the percentage normalized signal of Repli-seq (***Hansen et al., 2010***), a high-throughput sequencing for quantifying DNA replication time as a function of genomic position, between the elusive and non-elusive genes. The results showed that elusive genes were prone to late replication in all of the 15 cell lines examined (***Figure 6c***, ***Figure 6—figure supplement 3***). Late-replicating regions are frequently located at the nuclear periphery and often interact with the nuclear lamina. Therefore, we examined the nuclear position of the genomic regions including the elusive genes by referring to the lamina associating domains (LADs) that were identified by the ChIP-seq reads for Lamin B1 (***van Schaik et al., 2020***; ***Zheng et al., 2018***). Compared with the non-elusive genes, the elusive genes were found to be enriched in LADs for all of the four cell lines examined (***Figure 6d***, ***Figure 6—figure supplement 4***), consistent with their late replication timings (***van Steensel and Belmont, 2017***).

We further investigated the association of the restricted expressions of the elusive genes with epigenetic features. From 739 elusive genes whose expressions were quantified in the GTEx

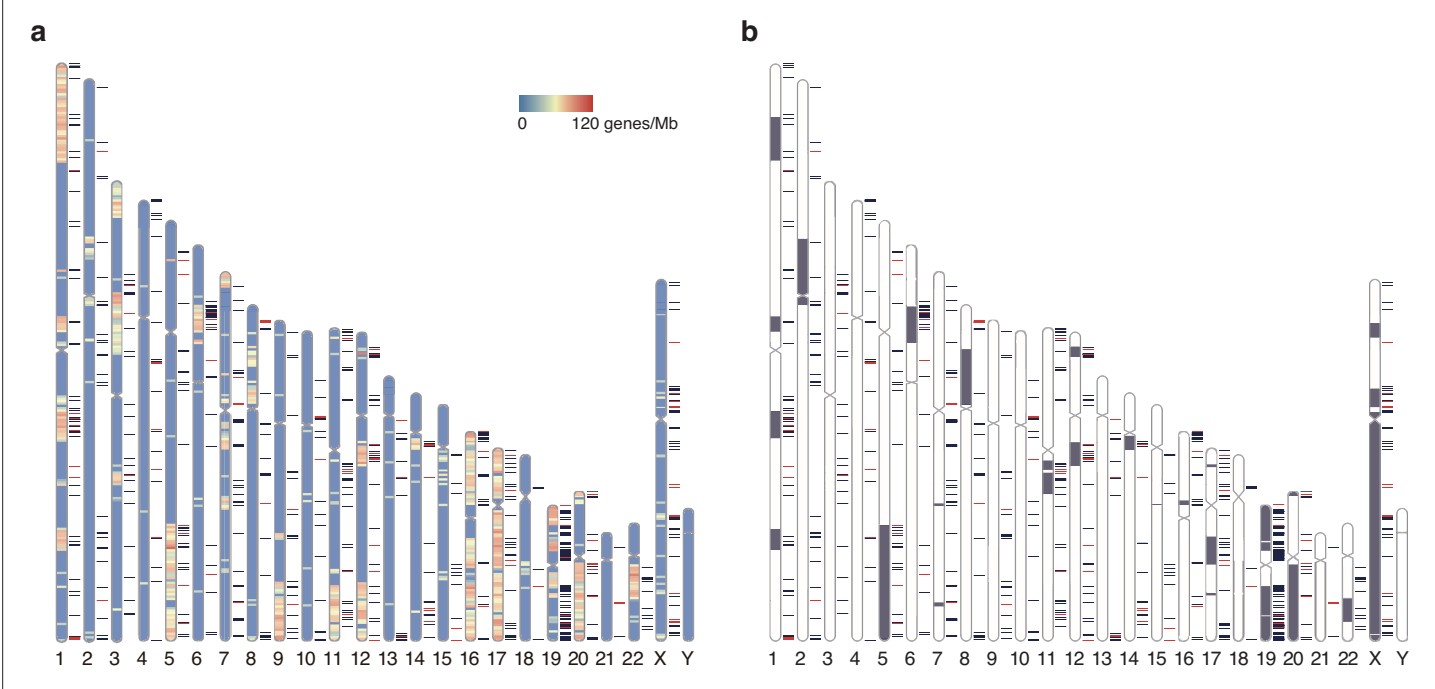

**Figure 7.** Chromosomal distribution of human elusive genes. Red and dark blue horizontal bars beside the chromosome ideogram represent the location of elusive genes with restricted expression (Shannon's $H' < 1$) and the other elusive genes, respectively. (**a**) The chromosome diagrams are colored according to the density of the genes that harbor chicken orthologs in microchromosomes (number of genes/Mb). 93 and 68 elusive genes were orthologous to the chicken genes in macro- and microchromosomes, respectively, and 4211 and 2078 non-elusive genes were orthologous to the chicken genes in macro- and microchromosomes, respectively. This indicates that the chicken orthologs of the elusive genes are abundant in microchromosomes compared with those of the non-elusive genes (p=0.0175). (**b**) Gray regions in the diagram indicate orthologous regions of microchromosomes in the ancestors of gnathostomes (*Nakatani et al., 2021*). 395 and 296 elusive genes were located in the genomic regions corresponding to ancient macro- and microchromosomes, respectively, and 5950 and 1929 non-elusive genes were located in the genomic regions corresponding to these ancient chromosomes. The result recapitulates the biased localization of the elusive genes on microchromosomes (p=9.50 × 10⁻²⁴). The chromosome diagrams were drawn using RIdeogram (*Hao et al., 2020*).

The online version of this article includes the following figure supplement(s) for figure 7:

**Figure supplement 1.** Distribution of elusive genes across human chromosomes.

database, we classified the elusive genes into two groups based on the pleiotropy in terms of gene expressions: that is, 589 elusive genes with Shannon's diversity index $H' \geq 1$ were ubiquitously expressed, that is, more pleiotropic, and 150 of those with $H' < 1$ were expressed in only a few or none of the tissues examined, that is, less pleiotropic (*Figure 5*). Importantly, all of the four epigenetic features of the elusive genes with $H' < 1$ were more pronounced than those with $H' \geq 1$: sparse ATAC-seq peaks, short TADs, late replication timings, and significant overlaps with LADs (*Figure 6*, *Figure 6—figure supplements 1–4*). This observation suggests that low-level and spatially restricted expressions of the elusive genes are associated with epigenetic features of these genomic regions.

High GC contents in genomic regions potentially hinder identifying an epigenetic feature by short-read sequencing because of the underrepresentation of sequence reads by amplification-based sequencing libraries. This bias might lead to sparse distributions of the ATAC-seq peaks and Hi-C contacts in the genomic regions that contain the elusive genes. However, only 3.00 and 9.00% of the elusive genes with $H' < 1$ and $H' \geq 1$ were located in regions of extremely high GC content (>60%), respectively, showing that the elusive genes $H' \geq 1$ rather tend to contain more genes with high GC content (p=0.0176). Thus, the depleted epigenomic features in the genomic regions containing elusive genes are unlikely to be false discoveries caused by a technical issue, namely the underrepresentation of the sequencing reads.

## Elusive gene orthologs in the chicken microchromosomes

The heterogeneous locations of the elusive genes can also be examined from a chromosome-scale viewpoint (*Figure 7*, *Figure 7—figure supplement 1*). The visualization via chromosome ideograms indicated an overlap of the elusive genes with the genomic regions enriched for the genes whose chicken orthologs are on the microchromosomes (chromosomes 11–38 and W), providing a statistical support for this trend (p=0.0175; *Figure 7a*). Indeed, microchromosomes of the chicken and other vertebrate exhibit genomic features including high GC content, high gene density, and rapid nucleotide substitutions in comparison with their macrochromosomes (*Groenen et al., 2009*; *International Chicken Genome Sequencing Consortium, 2004*; *Schield et al., 2019*; *Waters et al., 2021*), which also characterize genomic regions containing elusive genes. On the contrary, previous studies revealed that the chicken microchromosomes are preferentially located in the A compartments of the nucleus (*Perry et al., 2020*) and are early replicating (*McQueen et al., 1998*). These characteristics associated with the microchromosomes were opposite characteristics to the human genomic regions preferentially containing the elusive genes.

We further analyzed the ATAC-seq peaks in the chicken genome and found more peaks in the genomic regions including the elusive gene orthologs than in those containing non-elusive gene orthologs in four samples out of eight and no significant differences in the peak density in the four remaining samples (*Figure 6—figure supplement 5*). These observations indicate that, in an epigenetic manner, the chicken orthologs of the elusive genes are not regulated to reduce their expression level. This idea was further supported by a comparison of the expression profiles between the chicken orthologs of the elusive and non-elusive genes, showing no significant differences between them (*Figure 5—figure supplement 1*). Our analyses indicate that the genomic features of the elusive genes such as high GC and high nucleotide substitutions do not always correlate with a reduction in pleiotropy of gene expression that potentially leads to an increase in functional dispensability in the course of vertebrate evolution. In addition, avian orthologs of the elusive genes did not show higher $K_A$ and $K_S$ values than those of the non-elusive genes (*Figure 3*, *Figure 2—figure supplement 1*), likely consistent with not significant difference in gene expression levels between them in the species (*Figure 5—figure supplement 1*; *Cherry, 2010*; *Zhang and Yang, 2015*). We further compared expression profiles between the orthologs of the human elusive and non-elusive genes in several non-mammalian vertebrates and found that the orthologs of the elusive genes tend to exhibit low pleiotropy in green anole, coelacanth, and gar but not in Western clawed frog. The result suggests that the low pleiotropy of the elusive genes has persisted at least since the bony vertebrate ancestors. With respect to the chicken genome, the 'elusive' features for the genes orthologous to human elusive genes might have been relaxed—functional importance of the orthologs has increased—during evolution leading to chicken.

## Discussion

Here we identified elusive genes that were lost in multiple lineages during mammalian evolution using a comprehensive scan of gene phylogenies. To identify gene loss events, absence of evidence (i.e. missing genes caused by incomplete genome assemblies and gene annotations) should be reviewed meticulously (*Deutekom et al., 2019*). Additionally, gene loss might be detected erroneously because of failure in similarity searches for orthologs of rapidly evolving genes (*Moyers and Zhang, 2015*). In this study, we aimed to reduce these false discoveries through our multifaceted approaches (*Figure 1*). We selected those species with highly complete gene annotations through integration of multiple gene annotations. Using these improved gene annotations, we created orthologous groups by employing a highly sensitive homology search with MMSeqs2 (*Steinegger and Söding, 2017*) and merged them into those identified in the Ensembl database. Furthermore, we restricted the loss events that were observed as gene absence in all species examined within all hierarchical levels of the selected taxonomic groups (*Figure 1b*). This absence is likely to have occurred as a gene loss in the common ancestor of the particular taxon rather than as a false discovery of gene loss in the individual species independently. Genuine continuous (e.g. telomere-to-telomere) genome assemblies are now available using modern sequencing technologies (*Nurk et al., 2022*). These genomic assemblies may help relieve the labor of examining for information losses, thereby facilitating the identification of genuine gene loss in any given species.

In the human genome, the elusive genes and their flanking regions harbor particular characteristics, including high GC content and high gene density, that may have originated long before the emergence of mammals (*Figure 3*). Frequent synonymous variations across modern humans in the elusive genes, consistent with higher synonymous substitution rates between the vertebrate orthologs, suggest that the genomic regions including elusive genes have been subject to rapid evolution for approximately 500 million years (*Figures 2 and 4*). Our findings indicate that heterogeneous genomic characteristics potentially affect the fate of genes at the latest period of vertebrate evolution. Analyses with large numbers of germline mutations in the human genome have illustrated the heterogeneity of mutation rates (*Campbell and Eichler, 2013*; *Seplyarskiy and Sunyaev, 2021*; *Terekhanova et al., 2017*). High GC content in the elusive genes may have facilitated an elevation of the mutation rate, as observed in the enrichment of rare variants in high-GC regions in the human genome (*Schaibley et al., 2013*). In addition, some of the elusive genes appear to have retained particular epigenetic marks including sparse ATAC-seq peaks, late replication timings, and location within LADs (*Figure 6—figure supplements 1–4*); these epigenetic marks are relevant to an increase in the mutation rate. Genomic regions with late replication timing exhibit increased mutation rates because of their unstable structure during the S-phase of the cell cycle (*Koren et al., 2012*; *Stamatoyannopoulos et al., 2009*). LADs retain more G-to-T mutations because of their susceptibility to oxidative damage in the nuclear periphery resulting in high levels of 8-oxoguanine (*Yoshihara et al., 2014*). Close coordination of the studies on gene evolution with germline mutation repertoires and spectra, which can be approximated from the collection of de novo mutations obtained by trio sequencing, may further facilitate our understanding of gene fates driven by heterogeneous genomic features—this would be viewed as 'mutation-driven' evolution (*Nei, 2013*).

The epigenetic marks of elusive genes are relevant to the suppression of gene expression (*van Steensel and Belmont, 2017*), and indeed, these genes harbor weakened and spatially restricted expression profiles (*Figures 5 and 6* and *Figure 6—figure supplements 1–4*). However, the genomic features associated with these epigenetic marks usually exhibit lower GC contents and reduced gene density (*Gilbert et al., 2004*; *Rao et al., 2014*; *van Steensel and Belmont, 2017*). This discrepancy may be caused in part by a gain of local heterochromatin accompanied by suppression of the expression of transposable elements, as observed in various eukaryotic genomes (*Choi and Lee, 2020*; *Fiston-Lavier et al., 2007*; *Grewal and Jia, 2007*; *Rangasamy, 2013*; *Slotkin and Martienssen, 2007*; *Underwood et al., 2017*). Previous analyses showed frequent heterochromatinization of the human genomic regions where KRAB zinc finger genes colocalize with L1 retrotransposons (*Imbeault et al., 2017*; *O'Geen et al., 2007*; *Vogel et al., 2006*). One of the genomic regions found in human chromosome region 19p12 also contains many elusive genes (*Vogel et al., 2006*; *Figure 7*). Closer attention to the local gene and repeat contents including repetitive elements and tandem gene clusters might facilitate our understanding of heterochromatinization in restricted genomic regions, although we excluded such gene clusters in our search for elusive genes (*Figure 1a*).

A chromosomal-scale view of the distribution of elusive genes illuminated their significant correlation with the genes whose chicken orthologs are located on microchromosomes (*Figure 7a*). More importantly, genomic regions rich in elusive genes were traced back to the microchromosomes of the ancestral gnathostomes by reconstructing chromosomes of the ancestral genomes (*Figure 7b*). This inference of ancestral karyotypes augments our observations that some elusive natures of genomic sequences have been retained for hundreds of millions of years (*Figure 3*). In other words, the result suggests that the disparity of genomic regions that allows the 'elusiveness' for the genes has been retained during vertebrate evolution. On the other hand, comparisons of the expression profiles between the orthologs of the elusive and non-elusive genes for non-mammalian vertebrates suggest that the orthologs of the elusive genes have been associated with a reduction in pleiotropy of gene expression since vertebrate ancestors but acquired the diverse expressions in chicken and frog (*Figure 5—figure supplement 1*). Additionally, in the chicken genome, the diverse expressions of the chicken orthologs of the human elusive genes may be correlated with the abundance of ATAC-seq peaks (*Figure 6—figure supplement 5*). These findings again suggest that the chicken orthologs of the human elusive genes have increased pleiotropy of gene expression, which may lead to a lineage-specific acquisition of functional indispensability. It should be noted that the choices of tissues used in these analyses were largely different between the human and non-mammalian vertebrates (Tables S3 and S4; *Supplementary file 1b and c*). The chicken ATAC-seq data could be obtained only from

developing embryos, while the human ATAC-seq in ENCODE were performed with cell lines. Therefore, the aforementioned interpretation should be treated carefully.

Finally, we note the potential evolutionary courses that facilitate the transition of gene fate from retention to loss. One possible course is a decrease in essential functions because of rapid sequence evolution in local genomic regions. The elusive genes located in those genomic regions with rapidly evolving characteristics are likely to accumulate neutral or even moderately harmful mutations in coding regions frequently, resulting in impaired essential functions. Another factor is the spatiotemporal suppression of gene expression via epigenetic constraints. Previous studies showed that lowly expressed genes are associated with low functional essentiality (*Cherry, 2010*; *Gout et al., 2010*), as shown for elusive genes in our study. Elusive genes with reduced pleiotropy may have limited opportunities to function, potentially leading to loss of their important roles. The extent of these evolutionary forces may have varied with time and lineages, resulting in a patchy loss of elusive genes phylogenetically. Interestingly, a recent large-scale scan of de novo mutations in *Arabidopsis* indicates the association of mutation rates with epigenetic features and functional essentiality of genes (*Monroe et al., 2022*). Further investigation of the association of genes with the surrounding genomic regions in various taxa may provide a common understanding of genomic and epigenomic features that potentially alter the fate of genes. Although epigenetic features are plastic, our findings indicate that the disparities of genomic regions are reflected in the heterogeneity of evolutionary forces and have been retained for hundreds of millions of years. This idea prompts us to explore evolutionary constraints on more global genomic regions that are potentially associated with structural characteristics including chromosomal composition and locations within the nucleus.

## Materials and methods

### Sequence retrieval

We retrieved genome assemblies and gene annotations of 114 mammals and 132 non-mammal vertebrates from RefSeq (accessed on April 9, 2018), Ensembl release 92, and the repositories of the individual genome projects (Supplementary Table S1 in *Supplementary file 1a*). Gene annotations for a single species from multiple repositories were integrated into one as follows. When gene annotations of multiple repositories were referring to the same version of the genome assembly, the annotation GTF files were merged with the 'cuffcompare' tool (*Trapnell et al., 2012*). Otherwise, translated amino acid sequences were clustered by CD-HIT v. 4.6.4 (*Fu et al., 2012*) with 100% sequence similarity, and the representative sequence for each cluster was retrieved by assuming that each cluster represented a single locus. Subsequently, we selected the canonical amino acid sequence for each locus: canonical peptides of the Ensembl genes were retrieved from the Ensembl database; for other resources, the longest amino acid sequence from the isoforms of a locus was chosen. The completeness of the gene annotations was performed on the gVolante web server with assessments by BUSCO v.2 (*Simão et al., 2015*) by referring to the vertebrate ortholog sets provided by BUSCO and CVG (*Hara et al., 2015*). The gene annotations of mammals, birds, and ray-finned fishes that had fewer than 1% missing genes, as well as those of the other vertebrates with fewer than 3% missing genes, were selected. Exceptionally, the gene annotations of *Gavialis gangeticus* (Reptilia; CVG missing ratio 3.86%), *Paroedura picta* (Reptilia; BUSCO vertebrate ortholog missing rate 3.25%), and *Scyliorhinus torazame* (Chondrichthyes; BUSCO vertebrate ortholog missing rate 4.45%) were added. Finally, the amino acid sequence set of 90 mammals and 101 non-mammalian vertebrates was subjected to t ortholog clustering. We also retrieved coding nucleotide sequences of the canonical amino acid sequences.

### Ortholog clustering and tree inference

We retrieved gene trees of human protein-coding genes and their homologs from Ensembl Gene Tree release 92. From these gene trees, we constructed an amino acid sequence set of the homologs consisting of the species selected in the above section. This sequence set, restricted to Ensembl sequences only, was used as the 'backbone' of the ortholog set of all the selected species. In addition, we generated ortholog groups for all the species used by employing OrthoFinder2 v. 2.3.3 (*Emms and Kelly, 2019*) based on the similarity of amino acid sequences: a sequence similarity search was performed using MMSeqs2 v. 2339462c06eab0bee64e4fc0ebebf7707f6e53fd (*Steinegger and*

*Söding, 2017*). The Ensembl and OrthoFinder ortholog sets were then merged to create the united set of ortholog groups, yielding 50,768 vertebrate ortholog groups.

The integrated ortholog groups were then subjected to molecular phylogenetic analysis. Amino acid sequences of the individual groups were aligned with MAFFT v. 7.402 (*Katoh and Standley, 2013*), and ambiguous alignment sites were removed with trimAl v1.4 (*Capella-Gutiérrez et al., 2009*). Phylogenetic trees were inferred with IQ-Tree v. 1.6.6 (*Nguyen et al., 2015*) by selecting the optimal amino acid substitution model with ModelFinder (*Kalyaanamoorthy et al., 2017*) implemented in the IQ-Tree tool for each sequence alignment. In the inferred phylogenetic trees, ambiguously bifurcated nodes—those with branch lengths less than 0.0025—were collapsed into a multifurcational node by the 'di2multi' function implemented in ape v. 5.5 (*Paradis and Schliep, 2019*). The trees were then rooted with the automatic rooting function 'get_age_balanced_outgroup' implemented in ete3 v. 3.1.1 (*Huerta-Cepas et al., 2016*) to minimize any discrepancy of tree topologies with the taxonomic hierarchy of the species included. Using the ortholog groups, the age of individual genes was estimated by inferring the oldest evolutionary lineage in the gene trees. We also adopted the gene age instructed by the Ensembl Gene Tree, wherever it shows an older age.

## Identification of elusive genes in the human genome

For the individual trees, orthologs of the human genes were detected by the 'get_my_evol_events' function in ete3 (*Huerta-Cepas et al., 2007*). This function inferred gene duplication nodes in the rooted trees, resulting in separation of the trees into 17,495 subtrees of mammalian ortholog groups containing human genes. The ortholog information was referenced to extract the species with no orthologs to humans. This absence was further assessed by the ortholog annotation of human genes in the Ensembl Gene Tree database.

We selected taxonomic groups for the individual mammalian ortholog groups in which the orthologs were missing in all the species examined (Table S1; *Supplementary file 1a*). We restricted our study to gene losses that were likely to have occurred in the common ancestor of particular taxonomic groups, rather than those arising from the incompleteness of gene annotations. When a gene was missing in all the taxonomic groups in the same hierarchy, we considered that the gene was lost in the common ancestor of these groups. Finally, we found 1233 human genes belonging to the ortholog groups that were absent in two or more taxonomic groups and defined them as elusive genes. The gene loss events inferred by molecular phylogeny were further assessed by synteny-based ortholog annotations implemented in RefSeq, as well as a homolog search in the genome assemblies (Table S1; *Supplementary file 1a*) with TBLASTN v2.11.0+ (*Altschul et al., 1997*) and MMSeqs2 (*Steinegger and Söding, 2017*) referring to the latest RefSeq gene annotations (last accessed on December 2, 2022). This procedure resulted in the identification of 813 elusive genes that harbored three or fewer duplicates. Similarly, we extracted 8050 human genes whose orthologs were found in all the mammalian species examined and defined them as non-elusive genes. Because these elusive and non-elusive genes were identified in the GRCh38 human genome assembly, we performed the following analyses using this assembly.

## Extraction of genomic and molecular evolutionary characteristics

We calculated the GC content of a gene by using its genomic region including introns and untranslated regions (UTRs). To calculate individual gene densities, we extracted genomic regions containing the genes and their flanking three genes at both ends and divided them by seven. The orthologs of the elusive and non-elusive genes were retrieved from the aforementioned gene trees. We computed $K_A$ and $K_S$ values of the ortholog pairs of human–chimpanzee, human–mouse, chicken–turkey (*Meleagris gallopavo*), central bearded dragon-green anole (*Anolis carolinensis*), and bamboo shark-whale shark (*Rhincodon typus*). To achieve this, we extracted ortholog groups that contained at least three of these ortholog pairs. Amino acid sequences of the human and the orthologs were aligned using MAFFT. Nucleotide sequence alignments of the coding regions were generated by 'back-translation' of the amino acid sequence alignments by trimAl, simultaneously removing ambiguous alignment sites. By employing coding nucleotide sequence alignments, numbers of synonymous and non-synonymous substitutions per site were computed using PAML v. 4.9a (*Yang, 2007*). To compute nucleotide sequence differences of the individual introns, we extracted 473 elusive and 4626 non-elusive genes that harbored introns aligned with the chimpanzee genome assembly. The nucleotide

differences were calculated via the whole genome alignments of hg38 and panTro6 retrieved from the UCSC genome browser.

## Multiomics analysis

Common and rare SNVs of the human populations were retrieved from dbSNP release 147 (*Sherry et al., 2001*), and human CNVs were obtained from the Database of Genomic Variants (DGV) release 2016-08-31 (*MacDonald et al., 2014*). The CNVs were classified into duplication and deletion variants, according to the annotation in DGV. The density of these variants in a gene was computed by dividing the number of variants identified in a gene region by its sequence length. *Z*-scores, indices of the tolerance against mutations, of synonymous, missense, and loss-of-function mutations of the individual genes were retrieved from gnomAD v. 2.1.1 (*Karczewski et al., 2021*).

Gene expression quantifications of adult and fetal tissues were retrieved from public databases. Expression profiles of adult tissues were obtained from the GTEx database v. 8 (*The GTEx Consortium et al., 2020*), computed by averaging TPM values across individuals. Expression profiles of fetal tissues were obtained from the Developmental Single Cell Atlas of gene Regulation and Expression (Descartes) portal (*Cao et al., 2020*) by calculating averaged TPM values of single cells. The maximum TPM values of the individual genes among the tissues were taken as the representative gene expression levels. As a proxy of the spatial diversity of gene expression, Shannon's species diversity index (*H′* values) was computed for each gene using the following equation:

$$H'_i = -\sum_{k=1}^{R} p_{i,k} ln p_{i,k}$$

where $H_i'$ represents the Shannon's index of *i*th gene in the list of the human genes, $p_{i,k}$ represents the proportion of the TPM values of the *i*th gene in the *k*th tissues/cell types, and *R* denotes the total number of tissues/cell types examined.

The ATAC-seq peaks and TAD boundaries of the human primary cells and culture strains were retrieved from the ENCODE 3 repository (Accession ID listed in Table S3; *Supplementary file 1b*; *The ENCODE Project Consortium, 2012*). Wavelet-smoothed signals of the ENCODE Repli-seq data were obtained from the UCSC genome browser (*Hansen et al., 2010*). The 20 kb bin-associated domains of LAD-seq that employed Lamin B1 antibodies (*van Schaik et al., 2020*) were retrieved from the 4D Nucleome Data Portal.

We also compared expression profiles and ATAC-seq peak densities between the orthologs of the elusive and non-elusive genes in non-mammalian vertebrates in a similar way as we did with the human datasets. Normalized gene expression profiles from RNA-seq data of normal adult tissues and early embryos for chicken, green anole, Western clawed frog, coelacanth, and spotted gar were obtained from the Bgee version 15 database (*Bastian et al., 2021*; Table S4; *Supplementary file 1c*). ATAC-seq narrow peak signals of chicken tissues were retrieved from NCBI GEO (Table S4; *Supplementary file 1c*) followed by coordination of the genome assembly with galGal5 with the UCSC lift-Over tool (*Hinrichs et al., 2006*) as needed.

## Code availability

The scripts for inferring gene presence and absence from gene trees were deposited in GitHub (https://github.com/yuichiroharajpn/ElusiveGenes, copy archived at *Hara, 2022*).

## Statistical tests

Comparisons of the genomic characteristics between the elusive and non-elusive genes were tested statistically with the nonparametric Mann–Whitney *U* test and Fisher's exact test implemented in R. Correction of multiple testing was performed using the Benjamini–Hochberg FDR approach. We considered $p < 0.05$ to be statistically significant.

## Acknowledgements

We would like to thank Yoichiro Nakatani for providing the information of orthologous regions of ancestral chromosomes in the human genome, and Hideya Kawaji and Ichiro Hiratani for insightful comments. We also would like to thank the peer reviewers for their constructive feedback and insightful comments, which have significantly improved this paper. This work was supported by RIKEN

to SK, JSPS KAKENHI under Grant Number 20H03269 to SK and 21K06132 to YH, AMED under Grant Number JP21wm0325050 to YH, and Mochida Memorial Foundation for Medical and Pharmaceutical Research to YH. Computations were partially performed on the NIG supercomputer at the ROIS National Institute of Genetics.

## Additional information

### Competing interests

Shigehiro Kuraku: Reviewing editor, eLife. The other author declares that no competing interests exist.

### Funding

| Funder | Grant reference number | Author |
|---|---|---|
| Japan Society for the Promotion of Science | KAKENHI | Yuichiro Hara |
| Japan Society for the Promotion of Science | KAKENHI | Shigehiro Kuraku |
| Mochida Memorial Foundation for Medical and Pharmaceutical Research | | Yuichiro Hara |
| Japan Society for the Promotion of Science | 21K06132 | Yuichiro Hara |
| Japan Society for the Promotion of Science | 20H03269 | Shigehiro Kuraku |
| Japan Agency for Medical Research and Development | JP21wm0325050 | Yuichiro Hara |

The funders had no role in study design, data collection and interpretation, or the decision to submit the work for publication.

### Author contributions

Yuichiro Hara, Conceptualization, Data curation, Funding acquisition, Writing – original draft, Writing – review and editing, Investigation, Methodology; Shigehiro Kuraku, Conceptualization, Supervision, Writing – review and editing, Funding acquisition

### Author ORCIDs

Yuichiro Hara ⓘ http://orcid.org/0000-0002-7817-8963
Shigehiro Kuraku ⓘ http://orcid.org/0000-0003-1464-8388

### Decision letter and Author response

Decision letter https://doi.org/10.7554/eLife.82290.sa1

## Additional files

### Supplementary files

• Supplementary file 1. Supplementary Tables S1, S3, S4. (a) Supplementary Table S1. Vertebrate species used for creating gene phylogenies. (b) Supplementary Table S3. ENCODE accession ID list used for epigenomic analyses. (c) Supplementary Table S4. RNA-seq and ATAC-seq samples of non-mammalian vertebrates.

• Supplementary file 2. Supplementary Table S2. Characteristics of the elusive and non-elusive genes in the human genome.

• MDAR checklist

## Data availability

The current manuscript is a computational study, so no data have been generated for this manuscript. Data from ENCODE Project was used and is available at https://www.encodeproject.org, IDs are shown in Table S3 in Supplementary File 1.

The following previously published dataset was used:

| Author(s) | Year | Dataset title | Dataset URL | Database and Identifier |
|---|---|---|---|---|
| van Schaik T, Vos M, Peric-Hupkes D, Hn Celie P, van Steensel B | 2020 | Dara from: Cell cycle dynamics of lamina-associated DNA | https://data.4dnucleome.org/publications/f1218a92-1f37-4519-85d6-ccedd5f7ad39 | 4D Nucleome Data Portal, f1218a92-1f37-4519-85d6-ccedd5f7ad39 |

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
