## [Editor Report]

The study provides a fundamental understanding of the driving forces behind gene losses in genome evolution and connects the propensity for gene losses to local genomic features like mutation rate and expression pattern. The methodology is compelling, as it identifies "elusive human genes" through independent gene losses in at least two mammalian lineages. The comparative genomics and statistical analyses are thorough and rigorous, making this study appealing to readers interested in exploring the global patterns and underlying mechanisms of gene fate evolution across the phylogenetic tree.

---

## [Decision Letter]

**Decision letter after peer review:**

Thank you for submitting your article "Gene fate spectrum as a reflection of local genomic properties" for consideration by *eLife*. Your article has been reviewed by 3 peer reviewers, including Wenfeng Qian as the Reviewing Editor and Reviewer #1, and the evaluation has been overseen by George Perry as the Senior Editor.

Essential revisions:

1) The identification of "elusive genes" in the current manuscript requires additional scrutinization, given it is the foundation of the whole study. Please check published pipelines in the identification of gene losses (e.g., TOGA – https://github.com/hillerlab/TOGA) and use additional tools such as BLASTX search to test for known technical artifacts when calling genes (homology detection failure, refer to https://journals.plos.org/plosbiology/article?id=10.1371/journal.pbio.3000862). Please give a reason for the parameters used in the analysis (e.g., CD-Hit clustering) or examine if the conclusions remain supported by various parameters in the computational pipeline. Also, please take a look at the enrichment of elusive genes in human chr19, and use the synteny-based age estimation of the elusive genes (Shao et al. 2019).

2) Please also present the features of other genes (the genomic background other than elusive and non-elusive genes). Are these genes show intermediate patterns between those of elusive and non-elusive genes? Please edit the manuscript accordingly if the definition of non-elusiveness is actually equivalent to the genomic background.

3) It would be informative to test the links between recombination rate / LD and the genomic locations of elusive genes (compared against randomly sampled genes).

4) Please control confounding factors such as gene expression level and confirm whether the proxy of mutation rate (i.e., Ks) is actually confounded by gene importance.

5) Please consider extending the analyses on fish and birds to other genomic features.

6) The authors should reconcile the findings in this study with previous reports about microchromosomes.

7) Please consider improving the clarity/presentation of Figures 5 and 7 and examine whether the pattern remains robust using various parameter sets.

8) Please think of a better term than "elusive gene" to describe the genes that were lost independently in different lineages. Please also clearly define other terms in the manuscript (e.g., functionally indispensable vs. importance, are they the same concept?)

9) Please consider presenting the current study in the framework of mutation-selection balance, and better explain the novelty of the study over previous tremendous studies about gene losses.

*Reviewer #1 (Recommendations for the authors):*

Line 18. "neutral factor" is better replaced by "factors independent of gene dispensability".

Line 47. "However" should be "on the contrary"?

Lines 113-114. As indicated in the weaknesses part, I am not fully convinced these genomic, epigenomic, and transcriptomic features are completely independent of gene function.

Figure 1b. Define the red and orange crosses in the legend.

Figure 7 appeared first in the Discussion section. Can this part be moved to the Results section?

*Reviewer #2 (Recommendations for the authors):*

Overall, I believe that this is an interesting study. However, this version of the manuscript could be significantly improved in terms of logical depth and methodological stringency.

1. Authors actually support the concept of mutation-driven evolution, i.e., the high mutation rate in genomic regions harboring the elusive genes would predispose their fate toward death. To increase the significance of their work, I suggest authors cite (Nei 2013; Xie et al. 2019) and put their work in a bigger context.

2. Authors mentioned that elusive genes are less important and thus more prone to loss. In my view, pleiotropy is a better term compared to importance. That is, elusive genes are less pleiotropic [e.g. narrowly expressed, Figure 5] and thus their loss are more tolerable or easily compensated by other genes. Actually, narrow expression breadth has been observed to be correlated with gene loss in both humans and flies (MacArthur et al. 2012; Yang et al. 2015).

3. I am generally convinced that authors reliably identified elusive genes by identifying gene loss events in the common ancestor of multiple descendant species (to control for errors induced by assembly or annotation, Figure 1). However, Figure 7 shows the enrichment of elusive genes in human chr19. This chromosome is well known to be enriched with tandemly duplicated Krueppel-associated box C2H2 zinc-finger protein family (KZNF), many of which are primate-specific (Shao et al. 2019). I suspect that tree-based strategy implemented in Figure 1 could not be able to dissect the evolution of this super complex gene family. I am proposing two specific analyses: how many elusive genes encoded by chr19 are KZNFs? how many of them have Ensembl one-to-one orthologs across mammals?

4. With the patterns in Figure 3 and 7, authors argued that features of elusive genes are deeply ancient and could be inherited from the microchromosomes of early vertebrates. This statement has multiple problems.

a) Figure 3 only show genomic level features (e.g., high GC content) conserved in multiple vertebrates including shark and chicken. Epigenetic features analyzed in Figure 5 to 6 were only based on human data. I suggest authors to extend these analyses to shark or chicken. Although some epigenetic data could not be available for these species, transcriptome data analyzed in Figure 5 should be available for at least some species.

b) In Figure 7, authors propose the concept related with microchromosomes. These chromosomes have been extensively studied, especially in birds. Some features of microchromosomes are consistent with that of elusive genes [e.g., high GC, (Bravo et al. 2021)]. However, microchromosomes are conserved in terms of gene order and their genes generally show high protein-level constraints as shown by low Ka/Ks (Waters et al. 2021; Li et al. 2022). Authors need to reconcile their discovery with the previous rich literature.

c) Line (L) 204, among 982 human elusive genes, only 540~390 are shared by other species (e.g., shark). I suggest taking advantage of genome-level synteny based age data generated in Shao et al. 2019 to examine the age distribution of human elusive genes. If a high proportion of them are dated as being old (e.g., shared by jawed vertebrates), the statement that these genes have an ancient origin could be better supported.

References

Bravo GA, Schmitt CJ, Edwards SV. 2021. What have we learned from the first 500 avian genomes. Annu Rev Ecol Evol Syst 52: 611-639.

Li M, Sun C, Xu N, Bian P, Tian X, Wang X, Wang Y, Jia X, Heller R, Wang M et al. 2022. de novo Assembly of 20 Chicken Genomes Reveals the Undetectable Phenomenon for Thousands of Core Genes on Microchromosomes and Subtelomeric Regions. Mol Biol Evol 39.

MacArthur DG, Balasubramanian S, Frankish A, Huang N, Morris J, Walter K, Jostins L, Habegger L, Pickrell JK, Montgomery SB et al. 2012. A systematic survey of loss-of-function variants in human protein-coding genes. Science 335: 823-828.

Nei M. 2013. Mutation-driven evolution. OUP Oxford.

Shao Y, Chen C, Shen H, He BZ, Yu D, Jiang S, Zhao S, Gao Z, Zhu Z, Chen X et al. 2019. GenTree, an integrated resource for analyzing the evolution and function of primate-specific coding genes. Genome Res 29: 682-696.

Waters PD, Patel HR, Ruiz-Herrera A, Alvarez-Gonzalez L, Lister NC, Simakov O, Ezaz T, Kaur P, Frere C, Grutzner F et al. 2021. Microchromosomes are building blocks of bird, reptile, and mammal chromosomes. Proc Natl Acad Sci U S A 118.

Xie KT, Wang G, Thompson AC, Wucherpfennig JI, Reimchen TE, MacColl AD, Schluter D, Bell MA, Vasquez KM, Kingsley DM. 2019. DNA fragility in the parallel evolution of pelvic reduction in stickleback fish. Science 363: 81-84.

Yang H, He BZ, Ma H, Tsaur SC, Ma C, Wu Y, Ting CT, Zhang YE. 2015. Expression profile and gene age jointly shaped the genome-wide distribution of premature termination codons in a *Drosophila melanogaster* population. Mol Biol Evol 32: 216-228.

*Reviewer #3 (Recommendations for the authors):*

– In recent years several gene loss pipelines were already published (e.g. TOGA – https://github.com/hillerlab/TOGA) and it would be highly beneficial for this study to compare their gene loss reports with output obtained from existing pipelines (which also address the false discovery rate issue)

– We did not appreciate Figure 5. We strongly recommend finding a more quantitative approach to visualise these results, since heat maps are misleading and show different x-axis and y-axis ranges (zoom-ins/ zoom-outs)

– All Figures and Supplementary Figures showing violin plots need to report the number of genes that underly these distributions.

[Editors' note: further revisions were suggested prior to acceptance, as described below.]

Thank you for resubmitting your work entitled "Gene fate spectrum as a reflection of local genomic properties" for further consideration by *eLife*. Your revised article has been evaluated by George Perry (Senior Editor) and a Reviewing Editor. A previous reviewer also read and commented on the revised manuscript.

Apparently, the manuscript has been significantly improved but there are some remaining issues that need to be addressed, as outlined below. In view of these comments, we kindly request that you consider revising the manuscript once more. Our hope is that, through this additional revision, the manuscript will be written more clearly and rigorously. Thank you for your understanding and continued efforts in improving your submission.

Reviewer #2 made the following comment. Please consider it during revision.

"The authors attempt to argue that the elusive status is ancient ("Thus, the heterogeneous genomic features driving gene fates toward loss have been in place since the ancestral vertebrates"). However, in response to my previous suggestion regarding chicken microchromosomes, the authors present mixed results. They observed high GC content, high gene density, and short gene length in chicken, similar to the findings in humans (Figure 3). Yet, the critical functional data between the two species are conflicting: human elusive genes exhibit low expression and fewer ATAC-seq peaks, while their chicken counterparts display the opposite pattern. In other words, chicken elusive genes exhibit higher pleiotropy, which may decrease the likelihood of their loss. Thus, these genes are not elusive, and high GC content, high gene density, and short gene length do not necessarily predict elusiveness. Given that the authors only analyzed the functional data of human and chicken genomes, it is not possible to determine whether the "elusive" status is ancient or derived from the human or mammalian lineage. I suggest that the authors analyze the transcriptome data in shark or spotted gar to provide further phylogenetic context. Otherwise, the authors should significantly tone down their statement."

The reviewing editor also has some comments on the title and abstract.

1. Please consider revising the title to "The Impact of Local Genomic Properties on the Evolutionary Fate of Genes Across Vertebrates".

2. Please consider adding a sentence (or something similar) at the end of the abstract. "This study sheds light on the complex interplay between gene function and local genomic properties in shaping gene evolution across vertebrate lineages."

---

## [Author Response]

Essential revisions:1) The identification of "elusive genes" in the current manuscript requires additional scrutinization, given it is the foundation of the whole study. Please check published pipelines in the identification of gene losses (e.g., TOGA – https://github.com/hillerlab/TOGA) and use additional tools such as BLASTX search to test for known technical artifacts when calling genes (homology detection failure, refer to https://journals.plos.org/plosbiology/article?id=10.1371/journal.pbio.3000862). Please give a reason for the parameters used in the analysis (e.g., CD-Hit clustering) or examine if the conclusions remain supported by various parameters in the computational pipeline. Also, please take a look at the enrichment of elusive genes in human chr19, and use the synteny-based age estimation of the elusive genes (Shao et al. 2019).

Based on this suggestion, we have repeated the analysis, and to more rigorously exclude possible false positives, and as a result, we have obtained a refined set of 813 elusive genes. The validation of gene loss has employed the ortholog annotations implemented in the RefSeq gene prediction pipeline, a synteny-based ortholog clustering platform other than TOGA, as well as similarity sequence search by MMSeqs2 that performs superior to NCBI BLAST.

For the revised manuscript, we have refined the elusive gene set as the reviewer suggested. In the genome assemblies, we have searched for the orthologs of the elusive genes for the species in which they were missing. The search has been conducted by querying amino acid sequences of the elusive genes with tblastn as well as MMSeqs2 that performed superior to tblastn in sensitivity and computational speed. In addition, regarding a comment by Reviewer 3 we have searched for the orthologs by referring to existing ortholog annotations. We used the ortholog annotations implemented in RefSeq instead of those from the TOGA pipeline: both employ synteny conservation. We have coordinated the identified orthologs with our gene loss criteria–absence from all the species used in a particular taxon–and excluded 268 genes from the original elusive gene set. These genes contain those missing in the previous gene annotations used in the original manuscript but present in the latest ones, as well as those falsely missing due to incorrect inference of gene trees. Finally, the refined set of 813 elusive genes were subject to comparisons with the non-elusive genes. Importantly, these comparisons retained the significantly different trends of the particular genomic, transcriptomic, and epigenomic features between them except for very few cases (Author response table 1). This indicates that both initial and revised sets of the elusive genes reflect the nature of the ‘elusiveness,’ though the initial set contained some noises. We have modified the numbers of elusive genes in the corresponding parts of the manuscript including figures and tables. Additionally, we have added the validation procedures in Methods.

**Author response table 1. sa2table1:** Difference in statistical significances across different elusive gene sets.

Features	Non-significant in theinitial gene set(1,081 elusive genes)	Non-significant in thecurrent gene set(**813**)	Non-significant in the current gene setexcluding chr19(669)
Gene density in the turkey genome	✓	✓	✓
Gene density in the green anole genome	✓		
Gene density in the bamboo shark genome	✓		
Gene density in the whale shark genome	✓	✓	✓
*K*_S_ in avians		✓	✓
*K*_A_ in avians	✓	✓	✓
*K*_A_ in sharks	✓	✓	✓
ATAC-seq peak density for GM23338	✓		
Lamin B1 ChIP-seq peak density for K562			✓

The other features showed significantly different trends between the elusive and non-elusive genes for all of the elusive gene sets and thus are not included in this table.

cd-hit clustering with 100% sequence identity only clusters those with identical (and sometimes truncated) sequences, and, in the cluster, the sequences other than the representative are discarded. This means that the sequences remain if they are not identical to the other ones. If the similarity threshold is lowered, both identical and highly similar sequences are clustered with each other, and more sequences are discarded. Therefore, our approach that employs clustering with 100% similarity may minimize false positive gene loss.

Please refer to the reply to [Rev2 point 3] for the abundance of the elusive genes on chromosome 19. To examine this point, we have performed the comparisons between the elusive and non-elusive genes excluding the genes on chromosome 19, and the characteristics remained unchanged even when chromosome 19 was excluded.

2) Please also present the features of other genes (the genomic background other than elusive and non-elusive genes). Are these genes show intermediate patterns between those of elusive and non-elusive genes? Please edit the manuscript accordingly if the definition of non-elusiveness is actually equivalent to the genomic background.

Our aim in this study is to extract the characteristics of the genes that differentiate their fates from retention to loss. To achieve this, we compared the genes with clearly different phylogenetic signatures for gene loss, namely elusive and non-elusive genes.

The remainders excluding the elusive and non-elusive genes do not necessarily exhibit intermediate features. Because our definitions of the elusive and non-elusive genes were stringent, the remainders may contain considerable numbers of genes with loss in more restricted taxa than our criterion. In addition, the reminders contain those with other particular phylogenetic signals (e.g., frequently duplicated). These do not necessarily exhibit intermediate features, and at least the former is rather closer to elusive genes.

3) It would be informative to test the links between recombination rate / LD and the genomic locations of elusive genes (compared against randomly sampled genes).

We have retrieved fine-scale recombination rate data of males and females from https://www.decode.com/addendum/ (Suppl. Data of Kong, A et al., Nature, 467:1099–1103, 2010) and have compared them between the gene regions of the elusive and non-elusive genes. Both comparisons show no significant differences: average 0.829 and 0.900 recombinations/kb for the elusive and non-elusive genes, respectively, *p*=0.898, for males; average 0.836 and 0.846 recombinations/kb for the elusive and non-elusive genes, respectively, *p*=0.256, for females.

4) Please control confounding factors such as gene expression level and confirm whether the proxy of mutation rate (i.e., Ks) is actually confounded by gene importance.

We thank the reviewer for this important comment. We totally agree that transcriptomic and epigenomic features cannot be easily distinguished from gene dispensability and do not think that these features of the elusive genes can be explained solely by intrinsic properties of the genomes. Our motivation for investigating the expression profiles of the elusive gene is to understand how they lost their functional indispensability (original manuscript L285-286 in Results). We also discussed the possibility that sequence composition and genomic location of elusive genes may be associated with epigenetic features for expression depression, which may result in a decrease of functional constraints (original manuscript L470-474 in Discussion). Nevertheless, we think that the original manuscript may have contained misleading wordings, and thus we have edited them to better convey our view that gene expression and epigenomic features are related to gene function.

(P.2, Introduction) “This evolutionary fate of a gene can also be affected by factors independent of gene dispensability, including the mutability of genomic positions, but such features have not been examined well.”

(P6, Introduction) “These data assisted us to understand how intrinsic genomic features may affect gene fate, leading to gene loss by decreasing the expression level and eventually relaxing the functional importance of ʻelusiveʼ genes.”

(P33, Discussion) “Another factor is the spatiotemporal suppression of gene expression via epigenetic constraints. Previous studies showed that lowly expressed genes reduce their functional dispensability (Cherry, 2010; Gout et al., 2010), and so do the elusive genes.”

Additionally, responding to the advices from Reviewers 1 and 2, we have added a new section Elusive gene orthologs in the chicken microchromosomes in which we describe the relationship between the elusive genes and chicken microchromosomes. In this section, we also argue for the relationship between the genomic feature of the elusive genes and their transcriptomic and epigenomic characteristics. In the chicken genome, elusive genes did not show reduced pleiotropy of gene expression nor the epigenetic features relevant with the reduction, consistently with the moderation of nucleotide substitution rates. This also suggests that the relaxation of the ‘elusiveness’ is associated with the increase of functional indispensability.

(P27, Elusive gene orthologs in the chicken microchromosomes in Results) “Our analyses indicates that the genomic features of the elusive genes such as high GC and high nucleotide substitutions do not always correlate with a reduction in pleiotropy of gene expression that potentially leads to an increase in functional dispensability, although these features have been well conserved across vertebrates. In addition, the avian orthologs of the elusive genes did not show higher *K*_A_ and *K*_S_ values than those of the non-elusive genes (Figure 3; Figure 3—figure supplement 1), likely consistent with similar expression levels between them (Figure 5—figure supplement 1) (Cherry, 2010; Zhang and Yang, 2015). With respect to the chicken genome, the sequence features of the elusive genes themselves might have been relaxed during evolution.”

Also, please refer to [Rev1- point2] for the mutability of the elusive genes. To examine this point, we have computed nucleotide sequence differences in introns, namely *K*_I_, between the human and chimpanzee genomes. This analysis revealed higher *K*_I_ values in the elusive genes than in the non-elusive genes, which is in line with our original hypothesis. The results have been added in the revised manuscript.

5) Please consider extending the analyses on fish and birds to other genomic features.

Please refer to the reply to [Rev2 point 4a] for the comparison between the elusive and non-elusive gene orthologs of nonmammalian vertebrates. We analyzed expression profiles and ATAC-seq peak densities of the elusive and non-elusive gene of these animals. We have created a new section *Elusive gene orthologs in the chicken microchromosomes* in Results and described the results in this section and Discussion.

We analyzed expression profiles and ATAC-seq peak densities of the elusive and non-elusive gene of these animals. We have created a new section *Elusive gene orthologs in the chicken microchromosomes* in Results and described the results in this section and Discussion.

We appreciate the reviewer for this meaningful suggestion. As a response, we have computed the differences in intron sequences between the human and chimpanzee genomes and compared them between the elusive and non-elusive genes. As expected, we found larger sequence differences in introns for the elusive genes than for the non-elusive genes. In Figure 2c of the revised manuscript, we have included the distribution of *K*_I,_ sequence differences in introns between the human and chimpanzee genomes for the elusive and non-elusive genes. Additionally, we have added the corresponding texts to Results and the procedure to Methods as shown below.

(P11, Identification of human ‘elusive’ genes in Results) “In addition, we computed nucleotide substitution rates for introns (*K*_I_) between human and chimpanzee (*Pan troglodytes*) orthologs and compared them between the elusive and non-elusive genes.”

(P11, Identification of human ‘elusive’ genes in Results) “Our analysis further illuminated larger *K*_S_ and *K*_I_ values for the elusive genes than in the non-elusive genes (Figure 2b, c; Figure 2—figure supplement 1). Importantly, the higher rate of synonymous and intronic nucleotide substitutions, which may not affect changes in amino acid residues, indicates that the elusive genes are also susceptible to genomic characteristics independent of selective constraints on gene functions.”

(P39, Methods) “To compute nucleotide sequence differences of the individual introns, we extracted 473 elusive and 4,626 non-elusive genes that harbored introns aligned with the chimpanzee genome assembly. The nucleotide differences were calculated via the whole genome alignments of hg38 and panTro6 retrieved from the UCSC genome browser.”

6) The authors should reconcile the findings in this study with previous reports about microchromosomes.

Please refer to the reply to [Rev2-point 4b] for the reconciliation of our findings of the elusive genes with the features of microchromosomes. In the human genome, we found a significant overlap between the elusive genes and the genes whose chicken orthologs are located on microchromosomes. Although the chicken microchromosomes shared some sequence features such as high GC content and high *K*_S_ values in common with the elusive genes in our sense but exhibited opposite transcriptomic and epigenetic trends. Although the result does not change the basis of our study in the human genome, it indicates that, in the course of evolution, genomic features of the elusive genes are not always associated with a reduction of pleiotropy of gene expression. The results have been described in the newly created section *Elusive gene orthologs in the chicken microchromosomes* in Results.

7) Please consider improving the clarity/presentation of Figures 5 and 7 and examine whether the pattern remains robust using various parameter sets.

Please refer to the replies to [Rev3- point 2] for this point. First, following this suggestion, we have conducted a statistical test to see whether the elusive genes contain more genes with restricted expression profiles (*H*’ < 1) than the non-elusive genes, and this trend was statistically supported. We have added the gene numbers of the individual categories and the result of this statistical tests to Figure 5. Therefore, we think the classification of the elusive genes based on the threshold (*H*’ = 1) is reasonable in Figure 7.

In addition, we have conducted statistical tests in a similar way with different thresholds for *H*’ (2, 3, and 0.5) and found that the pattern remains robust (*p* = 3.85x10^-67^, 1.29x10^-50^, and 9.40x10^-46^ setting thresholds *H*’ at 2, 3, and 0.5, respectively, for the GTEx dataset and *p* = 8.35x10^-61^, 2.01x10^61^, and 1.25x10^-24^ setting thresholds *H*’ at 2, 3, and 0.5, respectively, for the Descartes dataset).

To use Figure 7 in a new section in Results, we have added an ideogram showing the distribution of the genes that retain the chicken orthologs in microchromosomes. In response to the comment by Reviewer 2 [Rev2- point 4b], we have performed statistical tests and found that the elusive genes were significantly more abundant in orthologs in microchromosomes than the non-elusive genes. Furthermore, the observation that the elusive genes prefer to be located in gene-rich regions was already statistically supported (Figure 2f).

As shown in Figure 5, Shannon’s *H'* ranged from zero to approximately 4 (exact maximum value is 3.97) and 5 (5.11) for the GTEx and Descartes gene expression datasets, respectively. Although the threshold *H'*=1 was an arbitrarily set, we think that it is reasonable to classify the genes with high pleiotropy from those with low pleiotropy.

8) Please think of a better term than "elusive gene" to describe the genes that were lost independently in different lineages. Please also clearly define other terms in the manuscript (e.g., functionally indispensable vs. importance, are they the same concept?)

The phrase ‘elusive gene’ was already used in our previous paper to follow the advice of peer-reviewers for our past manuscripts, and for consistency, we would like use this term, while we have modified the sentence introducing this term to more carefully explain it.

9) Please consider presenting the current study in the framework of mutation-selection balance, and better explain the novelty of the study over previous tremendous studies about gene losses.

Please refer to the replies below to convey our novelty. We have cited existing literature in the revised manuscript.

Reviewer #2 (Recommendations for the authors): Overall, I believe that this is an interesting study. However, this version of the manuscript could be significantly improved in terms of logical depth and methodological stringency.1. Authors actually support the concept of mutation-driven evolution, i.e., the high mutation rate in genomic regions harboring the elusive genes would predispose their fate toward death. To increase the significance of their work, I suggest authors cite (Nei 2013; Xie et al. 2019) and put their work in a bigger context.

We appreciate the reviewer for considering a way to enhance the significance of our study. We now recognize the importance of the study by Xie et al. (2019) reporting the recurrent loss of an enhancer that modulates fin morphogenesis in stickleback. As suggested, in the revised manuscript, we have cited this paper in Introduction.

(P4, Introduction) “In the stickleback genome, a *Pitx1* enhancer was independently lost in multiple lineages inhabiting freshwater due to its genomic location in a structurally fragile site, leading to recurrent loss of pelvic fins (Xie et al., 2019). Genes and genomic elements in such particular regions may be prone to loss in a more neutral manner than the relaxation of functional importance or via functional adaptations.”

Additionally, to enhance the broad interests of our study, we have cited the Dr. Nei’s book in Discussion as shown below.

(P31, Discussion) “Close coordination of the studies on gene evolution with germline mutation repertoires and spectra, which can be approximated from the collection of de novo mutations obtained by trio sequencing, may further facilitate our understanding of gene fates driven by heterogeneous genomic features—this would be viewed as ‘mutation-driven’ evolution (Nei, 2013).”

2. Authors mentioned that elusive genes are less important and thus more prone to loss. In my view, pleiotropy is a better term compared to importance. That is, elusive genes are less pleiotropic [e.g. narrowly expressed, Figure 5] and thus their loss are more tolerable or easily compensated by other genes. Actually, narrow expression breadth has been observed to be correlated with gene loss in both humans and flies (MacArthur et al. 2012; Yang et al. 2015).

We thank the reviewer for the thoughtful suggestion. We agree that the word pleiotropy is more suitable in our manuscript. We have modified the manuscript as shown below.

(P20, Transcriptomic natures of elusive genes in Results) “Our findings demonstrate that some elusive genes harbor low-level and spatially-restricted expression profiles, i.e., less pleiotropic states, which are rarely observed in the non-elusive genes.”

(P23, Epigenetic nature of elusive genes in Results) “… we classified the elusive genes into two groups based on the pleiotropy in terms of gene expressions: that is, 589 elusive genes with Shannon’s diversity index *H*¢ ≥ 1 were ubiquitously expressed, i.e, more pleiotropic, and 150 of those with *H*¢ < 1 were expressed in only a few or none of the tissues examined, i.e., less pleiotropic (Figure 5).”

(P34, Discussion) “Elusive genes with reduced pleiotropy may have limited opportunities to function, potentially leading to loss of their important roles.”

3. I am generally convinced that authors reliably identified elusive genes by identifying gene loss events in the common ancestor of multiple descendant species (to control for errors induced by assembly or annotation, Figure 1). However, Figure 7 shows the enrichment of elusive genes in human chr19. This chromosome is well known to be enriched with tandemly duplicated Krueppel-associated box C2H2 zinc-finger protein family (KZNF), many of which are primate-specific (Shao et al. 2019). I suspect that tree-based strategy implemented in Figure 1 could not be able to dissect the evolution of this super complex gene family. I am proposing two specific analyses: how many elusive genes encoded by chr19 are KZNFs? how many of them have Ensembl one-to-one orthologs across mammals?

Responding to this comment, we investigated the KZNF genes on chromosome 19. We identified 206 those genes in chromosome 19, of which 75 were found to be elusive. Of these, 30 genes retained one-to-one orthologs of the mouse or dog. Although we excluded the nearly identical paralogs in our pipeline in the original manuscript and have refined the elusive gene set with synteny-based ortholog annotation in the current manuscript, the elusive KZNF genes have remained in this refined gene set. The elusive KZNF genes on chromosome 19 were lost in some mammalian lineages and duplicated in early primates.

Motivated by this comment, we conceived a possibility whether the enrichment of the paralogs of the elusive genes on chromosome 19 overrepresents the features relevant to the ‘elusiveness’. We thus have created another set of elusive genes, those excluding the genes on chromosome 19, and performed comparisons on the genomic, transcriptomic, and epigenomic features of the elusive and non-elusive genes. The results showed that the significant/non-significant differences have been maintained except for a few cases (see Author response table 1), indicating that the characteristics remained unchanged even when chromosome 19 is excluded.

4. With the patterns in Figure 3 and 7, authors argued that features of elusive genes are deeply ancient and could be inherited from the microchromosomes of early vertebrates. This statement has multiple problems.a) Figure 3 only show genomic level features (e.g., high GC content) conserved in multiple vertebrates including shark and chicken. Epigenetic features analyzed in Figure 5 to 6 were only based on human data. I suggest authors to extend these analyses to shark or chicken. Although some epigenetic data could not be available for these species, transcriptome data analyzed in Figure 5 should be available for at least some species.

In response to this reviewer’s comment, we have investigated the transcriptomic and epigenetic characteristics of the orthologs of the elusive genes in non-mammalian vertebrates as we had done for human in the original manuscript. We have retrieved gene expression profiles of normal tissues and early embryos from bulk RNA-seq data of chicken, tropical clawed frog, coelacanth, and spotted gar, respectively, from the Bgee database (https://bgee.org/). We have then compared expression profiles between the orthologs of the elusive and non-elusive genes for the individual species using the same procedure as those in the manuscript. Only the anole, coelacanth, and gar orthologs of the elusive genes show the enrichment of low *H’* values (newly created figure in Figure 5—figure supplement 1). This indicates that the low pleiotropy of expression of the elusive genes is not always observed in the non-mammalian species.

Furthermore, we have compared epigenomic properties between the orthologs of the elusive and non-elusive genes in the chicken genome. We have retrieved ATAC-seq narrow peak data for tissues of chicken embryos from NCBI GEO and compared the density of ATAC peaks between the orthologs of the elusive and non-elusive genes. The result indicates that, in five samples out of the eight, the orthologs of the elusive genes retained more ATAC peaks than those of the non-elusive genes, and that the reminders did not show this difference (newly created figure in Figure 6—figure supplement 5). This observation may remind us of a link between the reduction of the ‘elusiveness’ and the decrease of functional dispensability in gene evolution. However, it should be interpreted carefully, as different sets of tissues were used for the transcriptomic and epigenomic analyses between human and chicken. As described above, the chicken ATAC-seq experiments were mainly performed with developing embryos, while the human ATAC-seq used in our study were performed with cell lines. Nevertheless, the cross-species comparison of the epigenetic features suggests that the sequence features relevant to the elusive genes do not always induce the epigenetic conditions for gene expression depletion.

We have newly created Figure 5—figure supplement 1 and Figure 6—figure supplement 5 as shown below, described the results in the newly created section Elusive gene orthologs in the chicken microchromosomes in Results, and discussed them in Discussion. Also, we described the procedures in Methods.

(P-26-27, Elusive gene orthologs in the chicken microchromosomes in Results)

“Elusive gene orthologs in the chicken microchromosomes

The heterogeneous locations of the elusive genes can also be examined from a chromosome-scale viewpoint (Figure 7; Figure 7—figure supplement 1). The visualization via chromosome ideograms indicated an overlap of the elusive genes with the genomic regions enriched for the genes whose chicken orthologs are on the microchromosomes (chromosomes 11-38 and W), providing a statistical support for this trend (*p*=0.0175; Figure 7a). Indeed, microchromosomes of the chicken and other vertebrate exhibit genomic features including high GC-content, high gene density, and rapid nucleotide substitutions in comparison with their macrochromosomes (Groenen et al., 2009; International Chicken Genome Sequencing Consortium, 2004; Schield et al., 2019; Waters et al., 2021), which also characterize genomic regions containing elusive genes. On the contrary, previous studies revealed that the chicken microchromosomes are preferentially located in the A compartments of the nucleus (Perry et al., 2020) and are early replicating (McQueen et al., 1998). These characteristics associated with the microchromosomes were opposite characteristics to the human genomic regions preferentially containing the elusive genes.

We further analyzed the ATAC-seq peaks in the chicken genome and found more peaks in the genomic regions including the elusive gene orthologs than in those containing non-elusive gene orthologs in four samples out of eight and no significant differences in the peak density in the four remaining samples (Figure 6; Figure 6—figure supplement 5). These observations indicate that, in an epigenetic manner, the chicken orthologs of the elusive genes are not regulated to reduce their expression level. This idea was further supported by a comparison of the expression profiles between the chicken orthologs of the elusive and non-elusive genes, showing no significant differences between them (Figure 5; Figure 5—figure supplement 1). Our analyses indicate that the genomic features of the elusive genes such as high GC and high nucleotide substitutions do not always correlate with a reduction in pleiotropy of gene expression that potentially leads to an increase in functional dispensability in the course of vertebrate evolution. In addition, avian orthologs of the elusive genes did not show higher *K*_A_ and *K*_S_ values than those of the non-elusive genes (Figure 3; Figure 3—figure supplement 1), likely consistent with not significant difference in gene expression levels between them in the species (Figure 5—figure supplement 1) (Cherry, 2010; Zhang and Yang, 2015). With respect to the chicken genome, the sequence features of the elusive genes might have been relaxed during evolution.”

(P32-33, Discussion) “A chromosomal-scale view of the distribution of elusive genes illuminated their significant correlation with the genes whose chicken orthologs are located on microchromosomes (Figure 7a). More importantly, genomic regions rich in elusive genes were traced back to the microchromosomes of the ancestral gnathostomes by reconstructing chromosomes of the ancestral genomes (Figure 7b). This inference of ancestral karyotypes augments our observations that some elusive natures of genomic sequences have been retained for hundreds of millions of years (Figure 3). In other words, the result suggests that the disparity of genomic regions which allows the ‘elusiveness’ for the genes has been retained during vertebrate evolution. On the other hand, comparisons of the expression profiles between the orthologs of the elusive and non-elusive genes for non-mammalian vertebrates indicate that the elusive genes are not always associated with the restricted expression profiles (Figure 5; Figure 5—figure supplement 1). Additionally, in the chicken genome, this trend in gene expression may be correlated with the abundance of ATAC-seq peaks in the elusive genes (Figure 6—figure supplement 5). These findings again suggest that the elusive genes are not always associated with a reduction in pleiotropy of gene expression, which may lead to an increase of functional dispensability during evolution. It should be noted that the choices of tissues used in these analyses were largely different between the human and non-mammalian vertebrates (Tables S3, S4). The chicken ATAC-seq data could be obtained only from developing embryos, while the human ATAC-seq in ENCODE were performed with cell lines. Therefore, the aforementioned interpretation should be treated carefully.”

(P40, Methods) “We also compared expression profiles and ATAC-seq peak densities between the orthologs of the elusive and non-elusive genes in nonmammalian vertebrates in a similar way as we did with the human datasets.

Normalized gene expression profiles from RNA-seq data of normal adult tissues and early embryos for chicken, green anole, Western clawed frog, coelacanth, and spotted gar were obtained from the Bgee version 15 database (Bastian et al., 2021) (Table S4). ATAC-seq narrow peak signals of chicken tissues were retrieved from NCBI GEO (Table S4) followed by coordination of the genome assembly with galGal5 with the UCSC liftOver tool (Hinrichs et al., 2006) as needed.”

b) In Figure 7, authors propose the concept related with microchromosomes. These chromosomes have been extensively studied, especially in birds. Some features of microchromosomes are consistent with that of elusive genes [e.g., high GC, (Bravo et al. 2021)]. However, microchromosomes are conserved in terms of gene order and their genes generally show high protein-level constraints as shown by low Ka/Ks (Waters et al. 2021; Li et al. 2022). Authors need to reconcile their discovery with the previous rich literature.

Responding to this reviewer’s comment, we have first examined whether the chicken orthologs of the elusive genes tend to be located on microchromosomes. We have classified the chicken orthologs of the elusive and non-elusive genes into those on macro- and microchrosomes. Fisher’s exact test has indicated that elusive genes are enriched in the chicken microchromomes (*p*=0.0175, odds ratio=1.46; Author response table 2). In addition, the elusive genes are enriched in the genomic regions corresponding to the ancestral microchromosomes in early vertebrates (*p*=9.50x10^-24^, odds ratio=2.31; Author response table 3).

**Author response table 2. sa2table2:** Number of chicken orthologs of elusive and non-elusive genes locating in macro- and microchromosomes.

	Elusive	Non-elusive
Macrochromosome (chr1-10, Z)	93	4211
Microchromosome (chr11-, W)	68	2078

Genes in non-chromosome scaffolds and mitochondrial genome were excluded Fisher's exact test *p*=0.0175, odds ratio=1.48.

**Author response table 3. sa2table3:** Number of elusive and non-elusive genes locating in the genomic regions derived from ancestral macro- and microchromosomes.

	Elusive	Non-elusive
Ancestral macrochromosome	395	5950
Ancestral microchromosome	296	1929

Genes in the genomic regions that did not correspond to the ancestral macro/microchromsomes were excluded. Fisher's exact test *p*=9.50x10^-24^, odds ratio=2.31.

As the reviewer noted, the genomic regions including the elusive genes share several genomic characteristics with microchromosomes: high GC content, high gene density, and high *K*_A_ and *K*_S_ values (although *K*_A_/*K*_S_ values are lower for newly identified genes in the chicken microchromosomes in Li et al., 2022). On the contrary, previous analyses showed that the chicken microchromosomes tend to be early replicating and located in the A compartment in the nucleus, where the genes are actively transcribed. These observations are concordant with our finding that the chicken orthologs of the elusive genes are rich in ATAC peaks (see in the previous reply). We could not determine which epigenomic states are ancestral, but the genomic features of high GC-content and high nucleotide substitutions are not always correlated to reduced pleiotropy of gene expression, potentially leading to an increase in functional dispensability.

In the revised manuscript, we have created a new section *Elusive gene orthologs in the chicken microchromosomes* in Results to describe the enrichment of the elusive genes in the chicken microchromsomes and the association of their genomic characters with those of the chicken orthologs, while we cite the literature mentioned by the reviewer. In addition, we have mentioned the relevance of the elusive genes to the ancestral macrochromosomes in early vertebrates in Discussion. The corresponding sentences in the revised manuscript have already been included above in the previous.

c) Line (L) 204, among 982 human elusive genes, only 540~390 are shared by other species (e.g., shark). I suggest taking advantage of genome-level synteny based age data generated in Shao et al. 2019 to examine the age distribution of human elusive genes. If a high proportion of them are dated as being old (e.g., shared by jawed vertebrates), the statement that these genes have an ancient origin could be better supported.

We thank the reviewer for an opportunity to revisit the age of the elusive genes. We have examined this for the revised manuscript. Though the reviewer suggested using GenTree (http://gentree.ioz.ac.cn/) for this purpose, this database consists of euteleostome animals and does not include chondrichthyans and more distantly related animals. Instead, we have used the ortholog groups of jawed vertebrates built in this study and further added older branching dates of their orthologs and paralogs from the Ensembl Gene Tree. Of the 813 elusive genes in the revised dataset, 152 retain only mammalian orthologs, and this proportion is higher than that of the non-elusive genes (65 out of 8,050, *p*=2.50x10^-110^). We have then extracted the 517 elusive and 7,900 non-elusive genes whose ancestors were dated at the early evolution of jawed vertebrates or older. This subset of old gene age allowed us to examine if the elusive genes also retain loss-prone nature in non-mammalian vertebrates. On average, 40% of the 517 elusive genes are found to be retained in the nonmammalian vertebrates. On the other hand, more than 90% of the 7,900 non-elusive genes are retained by these species. These results indicate that while relatively young genes are frequently found in the elusive genes, the loss-prone nature of the elusive genes in the non-mammalian vertebrates is recapitulated by using this gene set of old origin. We have replaced Figure 3—figure supplement 1 with a revised version.

(P14, Tracing elusiveness back along the vertebrate evolutionary tree in Results) “We found that 152 out of 813 elusive genes originated in mammalian lineages, and this proportion was larger than those of the elusive genes (65 out of 8050, *p* = 2.50 ´ 10^-110^), indicating that the elusive genes are more abundant in recently born genes than non-elusive genes. We then selected 517 elusive and 7,900 non-elusive genes that originated in the common ancestors of jawed vertebrates or earlier. These subsets allowed us to examine the degree of retention of non-mammalian vertebrate orthologs in the elusive and non-elusive genes. On average, approximately 40% of these elusive genes were found to be retained by non-mammalian vertebrates, while this proportion increased up to 90% for the non-elusive genes. (Figure 3—figure supplement 1a). In the coelacanth, gar, and shark, the orthologs of the elusive genes were less frequently retained by all the species than those of the non-elusive ones (Figure 3—figure supplement 1b). The results suggest that the origins of the loss-prone propensity of the elusive genes potentially date back to the period long before the emergence of the Mammalia.”

Reviewer #3 (Recommendations for the authors):– In recent years several gene loss pipelines were already published (e.g. TOGA – https://github.com/hillerlab/TOGA) and it would be highly beneficial for this study to compare their gene loss reports with output obtained from existing pipelines (which also address the false discovery rate issue)

Please refer to the reply to Essential revisions 1. We understand that TOGA is an ortholog search pipeline, whose output include gene loss information, employing synteny information. TOGA inputs genome alignments between the reference and all the target species, which requires tremendous computational time in our case involving a number of species. Instead, we have incorporated ortholog annotations implemented in RefSeq, another synteny-based ortholog detection platform, into our validation for gene loss.

– We did not appreciate Figure 5. We strongly recommend finding a more quantitative approach to visualise these results, since heat maps are misleading and show different x-axis and y-axis ranges (zoom-ins/ zoom-outs)

For the visibility of Figure 5, we have added the percentage of the genes of *H'*≥1 and *H'*<1 in the plots and performed the statistical test to see the fraction of the genes of *H'*≥1 and *H'*<1 is equal between the elusive and non-elusive genes. Additionally, we have coordinated x-axis and y-axis ranges between the elusive and non-elusive genes. We have replaced Figure 5 with a revised version containing these modified figure panels.

– All Figures and Supplementary Figures showing violin plots need to report the number of genes that underly these distributions.

We have added the sample numbers of the plots to all figures or figure legends.

[Editors' note: further revisions were suggested prior to acceptance, as described below.]

Apparently, the manuscript has been significantly improved but there are some remaining issues that need to be addressed, as outlined below. In view of these comments, we kindly request that you consider revising the manuscript once more. Our hope is that, through this additional revision, the manuscript will be written more clearly and rigorously. Thank you for your understanding and continued efforts in improving your submission.Reviewer #2 made the following comment. Please consider it during revision."The authors attempt to argue that the elusive status is ancient ("Thus, the heterogeneous genomic features driving gene fates toward loss have been in place since the ancestral vertebrates"). However, in response to my previous suggestion regarding chicken microchromosomes, the authors present mixed results. They observed high GC content, high gene density, and short gene length in chicken, similar to the findings in humans (Figure 3). Yet, the critical functional data between the two species are conflicting: human elusive genes exhibit low expression and fewer ATAC-seq peaks, while their chicken counterparts display the opposite pattern. In other words, chicken elusive genes exhibit higher pleiotropy, which may decrease the likelihood of their loss. Thus, these genes are not elusive, and high GC content, high gene density, and short gene length do not necessarily predict elusiveness. Given that the authors only analyzed the functional data of human and chicken genomes, it is not possible to determine whether the "elusive" status is ancient or derived from the human or mammalian lineage. I suggest that the authors analyze the transcriptome data in shark or spotted gar to provide further phylogenetic context. Otherwise, the authors should significantly tone down their statement."

We are aware of your concern about the inconsistency of the results between human and chicken, and therefore already included comparisons of the gene expression profiles between the orthologs of the elusive and non-elusive genes with several non-mammalian vertebrates in the previous revision (Figure 5—figure supplement 1). This comparison indicated that the orthologs of the elusive genes were rich in those with lower pleiotropy in anole, coelacanth, and gar, while this tendency was not found in chicken and frog. The result suggests that the orthologs of the elusive genes are likely to have retained the “elusive” features in the common ancestor of bony vertebrates.

In addition, regarding the reviewer’s comment, we should have clearly stated that in non-mammalians, the genes we focused on are not necessarily ‘elusive genes’ but rather ‘the orthologs of the human elusive genes’ in the previous manuscript. These orthologs do not always retain the ‘elusiveness’ as the human genes do. Interestingly, the orthologs of the elusive genes in the chicken genome do not exhibit significant differences in *K*_A_ and *K*_S_ values from those of the non-elusive genes, which is potentially associated with increased pleiotropy of gene expression in these genes. These observations suggest that the orthologs of the human elusive genes we identified have increased functional importance in the lineage leading to chicken.

We have revised the following parts of Results and Discussion to describe transcriptomic characteristics of the orthologs of the human elusive genes in non-mammalians more clearly in the current manuscript as included below.

Elusive gene orthologs in the chicken microchromosomes in Results

“We further compared expression profiles between the orthologs of the human elusive and non-elusive genes in several non-mammalian vertebrates and found that the orthologs of the elusive genes tend to exhibit low pleiotropy in green anole, coelacanth, and gar but not in Western clawed frog. The result suggests that the low pleiotropy of the elusive genes has persisted at least since the bony vertebrate ancestors. With respect to the chicken genome, the ‘elusive’ features the genes orthologous to human elusive genes might have been relaxed —functional importance of the orthologs have increased—during evolution leading to chicken.”

Discussion

“On the other hand, comparisons of the expression profiles between the orthologs of the elusive and non-elusive genes for non-mammalian vertebrates suggest that the orthologs of the elusive genes have been associated with a reduction in pleiotropy of gene expression since vertebrate ancestors but acquired the diverse expressions in chicken and frog (Figure 5; Figure 5—figure supplement 1). Additionally, in the chicken genome, the diverse expressions of the chicken orthologs of the human elusive genes may be correlated with the abundance of ATAC-seq peaks (Figure 6—figure supplement 5). These findings again suggest that the chicken orthologs of the human elusive genes have increased pleiotropy of gene expression, which may lead to a lineage-specific acquisition of functional indispensability.”

The reviewing editor also has some comments on the title and abstract.1. Please consider revising the title to "The Impact of Local Genomic Properties on the Evolutionary Fate of Genes Across Vertebrates".

Thank you for your suggestion. We would like to use the title that you suggested without the last two words (‘The Impact of Local Genomic Properties on the Evolutionary Fate of Genes’). This title is more likely to increase the accessibility of our paper to a broad readership beyond those who are primarily investigating vertebrates, potentially prompting them to investigate genomic properties associated with the fate of genes in diverse organisms such as invertebrates, fungi, and plants.

2. Please consider adding a sentence (or something similar) at the end of the abstract. "This study sheds light on the complex interplay between gene function and local genomic properties in shaping gene evolution across vertebrate lineages."

Thank you for your suggestion that strengthens our focus and conclusion. We have revised the abstract as follows.

Thus, the heterogeneous genomic features driving gene fates toward loss have been in place and may sometimes have relaxed the functional indispensability of such genes. This study sheds light on the complex interplay between gene function and local genomic properties in shaping gene evolution that has persisted since the vertebrate ancestor.